# *NOTCH1* gene amplification promotes expansion of Cancer Associated Fibroblast populations in human skin

Atul Katarkar [1,7], Giulia Bottoni [2,3,7], Andrea Clocchiatti[2,3], Sandro Goruppi [2,3], Pino Bordignon [1], Francesca Lazzaroni [1], Ilaria Gregnanin[4], Paola Ostano [4], Victor Neel[5] & G. Paolo Dotto [1,2,3,6 ✉]

Cancer associated fibroblasts (CAFs) are a key component of the tumor microenvironment. Genomic alterations in these cells remain a point of contention. We report that CAFs from skin squamous cell carcinomas (SCCs) display chromosomal alterations, with heterogeneous *NOTCH1* gene amplification and overexpression that also occur, to a lesser extent, in dermal fibroblasts of apparently unaffected skin. The fraction of the latter cells harboring *NOTCH1* amplification is expanded by chronic UVA exposure, to which CAFs are resistant. The advantage conferred by *NOTCH1* amplification and overexpression can be explained by NOTCH1 ability to block the DNA damage response (DDR) and ensuing growth arrest through suppression of ATM-FOXO3a association and downstream signaling cascade. In an orthotopic model of skin SCC, genetic or pharmacological inhibition of *NOTCH1* activity suppresses cancer/stromal cells expansion. Here we show that *NOTCH1* gene amplification and increased expression in CAFs are an attractive target for stroma-focused anti-cancer intervention.

[1] Department of Biochemistry, University of Lausanne, 1066 Epalinges, Switzerland. [2] Cutaneous Biology Research Center, Massachusetts General Hospital, Charlestown, MA 02129, USA. [3] Department of Dermatology, Harvard Medical School, Boston, MA 02125, USA. [4] Cancer Genomics Laboratory, Edo and Elvo Tempia Valenta Foundation, Biella 13900, Italy. [5] Department of Dermatology, Massachusetts General Hospital, Boston, MA 02114, USA. [6] International Cancer Prevention Institute, 1066 Epalinges, Switzerland. [7] These authors contributed equally: Atul Katarkar, Giulia Bottoni. ✉email: paolo.dotto@unil.ch

Cancer associated fibroblasts (CAFs) are a key component of the tumor microenvironment and play a central role in cancer initiation, progression, and metastasis[1]. In the skin, conversion of dermal fibroblasts into CAFs can drive the development of keratinocyte tumors and field cancerization, a major clinical condition characterized by multifocal and recurrent epithelial tumors associated with widespread changes of the surrounding stroma[2,3]. While frequent gene mutations with growth-promoting function have been found even in apparently normal epithelium[4], whether genetic changes occur also in underlying stroma remains to be addressed.

Multiple signaling pathways converge onto CAFs activation through a variety of epigenetic control mechanisms[1,5,6]. In addition, several studies have reported chromosome and gene copy number alterations in CAFs derived from breast, prostate, colorectal, and ovarian cancer[7–9]. However, these findings could not be confirmed by others who raised the issue of technical artifacts[10–12]. Besides different methods of analysis and samples, possible genetic changes in stromal fibroblasts need to be investigated in the context of different cancer types and stromal cell heterogeneity[13]. In this regard, studies on genomic integrity in skin CAF populations are important to conduct, given the persistent exposure of the skin to exogenous clastogenic agents such as UVA, which reaches the dermal cell compartment due to its high penetrating power.

NOTCH signaling is an evolutionary conserved pathway with a key role in cell proliferation, survival, and differentiation. Upon NOTCH receptor activation by proteolytic cleavage, the NOTCH intracellular domain (NICD) translocates into the nucleus where it binds to CSL (RBP-Jκ), converting it from a repressor into an activator of transcription[14–16]. As a result, CSL loss and NOTCH activation can exert a similarity of effects. This is the case in dermal fibroblasts, in which down-modulation of CSL expression, as it can be caused by UVA exposure[17], or NOTCH activation, by ligand-producing neighboring cells, induce an overlapping program of CAF-effector genes[3].

Both CSL and NOTCH have been shown to display independent functions[15,18]. We have recently unveiled a role of CSL in human dermal fibroblasts (HDFs) and CAFs, separate from gene transcription, as an essential component of a telomere binding/protective complex. CSL loss or down-modulation in HDFs and CAFs results in DNA damage and genomic instability[19]. Here we show that in CAFs the *NOTCH1* gene is frequently amplified and overexpressed, preventing the DNA damage response (DDR) through ATM association and suppression of downstream signaling. Sustained NOTCH1 expression is required for CAFs proliferation and expansion and provides a target of translational significance of stroma-focused anti-cancer intervention.

## Results

### CAFs display genomic aberrations with frequent *NOTCH1* gene amplification and increased expression.

We recently found that stromal fibroblasts associated with premalignant and malignant skin squamous cell carcinoma (SCC) lesions are characterized by increased genomic instability[19]. To assess whether the latter is associated with chromosomal rearrangements, we analyzed three independent CAF strains (at 2nd passage after tumor dissociation) by comparative genomic hybridization arrays (aCGH). Multiple genomic aberrations were found in CAFs, with a restricted number of common gene amplifications (Fig. 1a and Supplementary Data 1). Among the regions with the highest number of aCGH positive probes was the one encompassing the *NOTCH1* gene, which maps at the end of chromosome 9q (9q34.3). Amplification of *NOTCH1* and two other genes detected by aCGH (*ERCC2* and *UVSSA*) was further validated by qPCR

analysis, using two other genes with no detectable copy number variations in the arrays, GAPDH and RPLP0, for normalization and as negative control, respectively, by the same approach as in refs. [20,21] (Fig. 1b). Similar *NOTCH1* gene amplifications were also detected by this approach in six additional CAF strains (Fig. 1c), suggesting the high frequency of the event.

Fluorescent in situ hybridization (FISH) provides a method of choice for quantification of gene copy number variations (CNVs) at the individual cell level[22]. For the present purpose, we used a fluorescently labeled probe spanning a 200-Kb genomic region of the *NOTCH1* locus together with a second probe for an independent region of chromosome 9 (9q21.3), where the gene is located. As shown in Fig. 1d, FISH analysis of the various CAF strains, mostly at 2nd (CAF 11, 12, 13) or 3rd (CAF 8, 14, 15, 16) passage after tumor dissociation (except for CAF9 and 10 at passage 6), showed heterogeneous cell populations, with most cells harboring increased copies of the *NOTCH1* gene (3–5), which were either separate from each other or juxtaposed on chromosome 9, as gene duplications (Fig. 1d). Parallel analysis of matched HDFs (m-HDFs), derived from flanking apparently unaffected skin of the same patients, showed an inverse pattern, with the great majority of cells with two *NOTCH1* copies and only a small fraction harboring extra copies, mostly gene duplications (Fig. 1d). No increase in *NOTCH1* copy number was observed in similarly cultured HDF strains derived from foreskin (f-HDFs), suggesting that sun-exposed areas can harbor populations of HDFs with CNVs without displaying a pathologic phenotype.

To rule out possible artifacts due to culturing conditions, the analysis was extended to the excised tissue samples from which the CAF and m-HDF strains studied above were derived. Fluorescence-guided laser capture micro-dissection (LCM) followed by FISH assays showed *NOTCH1* gene amplification in the analyzed CAFs. Similar amplifications were also found in a restricted fraction of fibroblasts captured from flanking apparently unaffected skin, confirming the findings obtained with cultured cells (Fig. 2a).

In agreement with the above findings, qPCR analysis of laser-captured CAFs showed *NOTCH1* gene amplification in all examined samples except one (SCC CAF14; Fig. 2b), in which significant levels of *NOTCH1* gene amplification were detected only in cultured CAFs (Fig. 1c). Gene amplification is often connected with increased expression. RT-qPCR analysis of the same LCM samples showed increased *NOTCH1* expression in the CAFs carrying the amplified gene, with parallel CSL down-regulation and increased HES1 levels (Fig. 2c), consistent with the negative regulation of CSL expression by increased NOTCH1 activity demonstrated further below, and the fact that HES1 is a common target of CSL repression and NOTCH1 activation[14]. The results were complemented by similar analysis of freshly derived CAFs versus matched HDFs from these and additional three patients' samples with similar statistically significant results (Fig. 2d). Immunofluorescence analysis of the cultured CAFs showed a higher fraction of cells with elevated cleaved nuclear NOTCH1 (ICN1) levels than in m-HDFs (Fig. 2e). Parallel immunoblot analysis confirmed increased ICN1 expression in CAFs relative to matched and foreskin-derived HDFs, used as outside controls (Fig. 2f and Supplementary Fig. 1b, c).

Thus, CAFs from skin SCCs consist of heterogeneous populations with *NOTCH1* gene amplification and increased expression, which are also present in a lesser fraction of fibroblasts of flanking unaffected skin.

### NOTCH1 up-regulation ensures sustained CAF effectors gene expression and proliferation.

A complex relationship exists

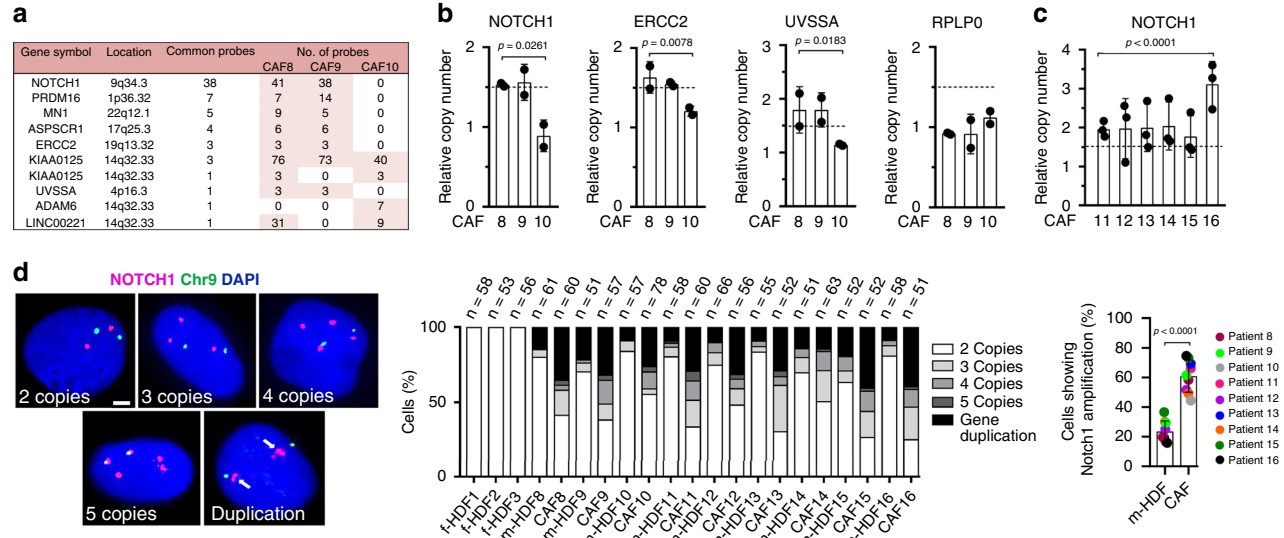

**Fig. 1 _NOTCH1_ gene amplification in SCC-derived CAFs. a** List of genes commonly amplified in at least 2 out of 3 CAF strains (CAF 8, 9, 10) as identified by Comparative Genomic Hybridization array (aCGH). For each gene, the chromosomal location, number of positive probes, and number of common probes are indicated. A complete list of copy number variations (CNVs) identified by aCGH of each CAF strain is provided in Supplementary Data 1. **b** Validation of _NOTCH1_, _ERCC2_, and _UVSSA_ gene amplifications in the same CAFs as in **a**, by qPCR analysis of two independent culture experiments (individual results indicated by dots) of the same strains. _GAPDH_ was used for the internal normalization and _RPLP0_ was used as negative control. DNA copies were calculated relative to three foreskin-derived HDF strains (f-HDF) as outside reference following the same approach as in refs. [20,21]. Data are presented as mean ± s.d. One sample _t_-test, *$p < 0.05$, **$p < 0.01$. n(CAF strain) = 3, n(f-HDF strain) = 3. **c** Quantification of _NOTCH1_ gene copy number in six additional CAF strains (CAFs 11-16) from three independent culture experiments of the same strains (individual results indicated by dots). DNA copies were calculated relative to three foreskin-derived HDF strains (f-HDF) strains as outside reference as in **b**. Data are presented as mean ± s.d. One sample _t_-test, ****$p < 0.0001$. n(CAF strain) = 6, n(f-HDF strain) = 3. **d** Representative images and quantification of percentage of cells with _NOTCH1_ gene copy number variations in 9 CAF strains as in **b** and **c**, their respective m-HDFs and three unmatched f-HDFs from healthy donors as assessed by FISH with a _NOTCH1_-specific probe (magenta) in parallel with a probe for _chr9_ q21.3 localization (green). Gene duplications are defined as multiple proximal _NOTCH1_ positive dots per chromosome. The number of analyzed nuclei (n) obtained from two independent experiments are shown on top of the corresponding bar. Overall quantification and statistical significance of percentage of cells with _NOTCH1_ amplifications in CAFs versus m-HDFs. Scale bar, 50 μm. Values for each strain are indicated as dots with mean ± s.d. Two-tailed paired _t_-test between m-HDF and CAF, ****$p < 0.0001$. n(CAF strains) = 9, n(m-HDF strains) = 9, n(f-HDF strains) = 3.

between NOTCH1 and CSL activity in HDFs and CAFs. CSL functions as a constitutive negative repressor of a large battery of CAF effector genes, which are all induced by decreased _CSL_ expression as it occurs at early steps of CAF activation[3,17,23,24]. Separately from its role in transcription, CSL is essential for maintenance of genomic stability as part of a telomere protective complex that is lost in CAFs[19]. CAF effector genes under negative CSL control can be induced by increased levels of activated NOTCH1, which, by binding to CSL, converts it from a repressor into an activator of transcription[3]. NOTCH1 activation can also suppress CSL expression as part of a negative feedback loop mediated by induction of HES/HEY family of transcriptional repressors[14].

In agreement with previous findings[3,19], silencing of the _CSL_ but not _NOTCH1_ gene in f-HDFs resulted in up-regulation of a number of CAF effector genes with a key tumor-promoting function (Fig. 3a and Supplementary Fig. 2a). Expression of all these genes was induced, while that of _CSL_ decreased, by enhanced NOTCH1 activity, by either lentiviral-mediated expression of ICN1 or ligand stimulation of the endogenous receptor (Fig. 3b and Supplementary Fig. 2b, d). Conversely, silencing of _NOTCH1_ in CAFs caused significant down-modulation of CAF effector genes, with similar changes elicited by treatment with a γ-secretase inhibitor (Deshydroxy LY-411575, DBZ) that suppresses endogenous NOTCH1 activation (Fig. 3c and Supplementary Fig. 2c). In many cellular systems, expression of JAGGED ligands is under positive NOTCH1 control as part of a self-reinforcing positive feedback

loop[14]. Even in f-HDFs, JAGGED 1 and 2 expression were induced by increased NOTCH1 activity (Fig. 3b and Supplementary Fig. 2b). In CAFs, JAGGED 1 and 2 expression was higher than in m-HDFs (Supplementary Fig. 2e) and suppressed by _NOTCH1_ silencing (Fig. 3c and Supplementary Fig. 2c).

As expected from previous findings[19], knockdown of _CSL_ in foreskin-derived HDFs resulted in DNA damage and growth suppression, which were not elicited by _NOTCH1_ silencing (Fig. 3d–f). In contrast to f-HDFs and m-HDFs, _NOTCH1_ silencing suppressed the proliferation of CAFs (Fig. 3g, left columns), without affecting the elevated levels of DNA damage resulting from loss of CSL in these cells[19] (Supplementary Fig. 2f, g), but inducing downstream events of the DNA Damage Response (DDR) shown further below. In contrast to foreskin-derived and matched HDFs, proliferation of CAFs was unaffected by UVA treatment (Fig. 3g, h), with inhibition of CAF proliferation by sustained _NOTCH1_ silencing being further decreased by UVA treatment (Fig. 3i). _NOTCH1_ silencing did not increase apoptosis in either HDFs or CAFs under basal conditions as well as after UVA exposure at the doses used for these experiments (Supplementary Fig. 2h).

As HDFs from unaffected skin of SCC patients contain a population with _NOTCH1_ gene amplification, we tested whether these cells have a proliferative advantage under conditions of persistent DNA damage as can result from repeated UVA exposure. To test this hypothesis, we established

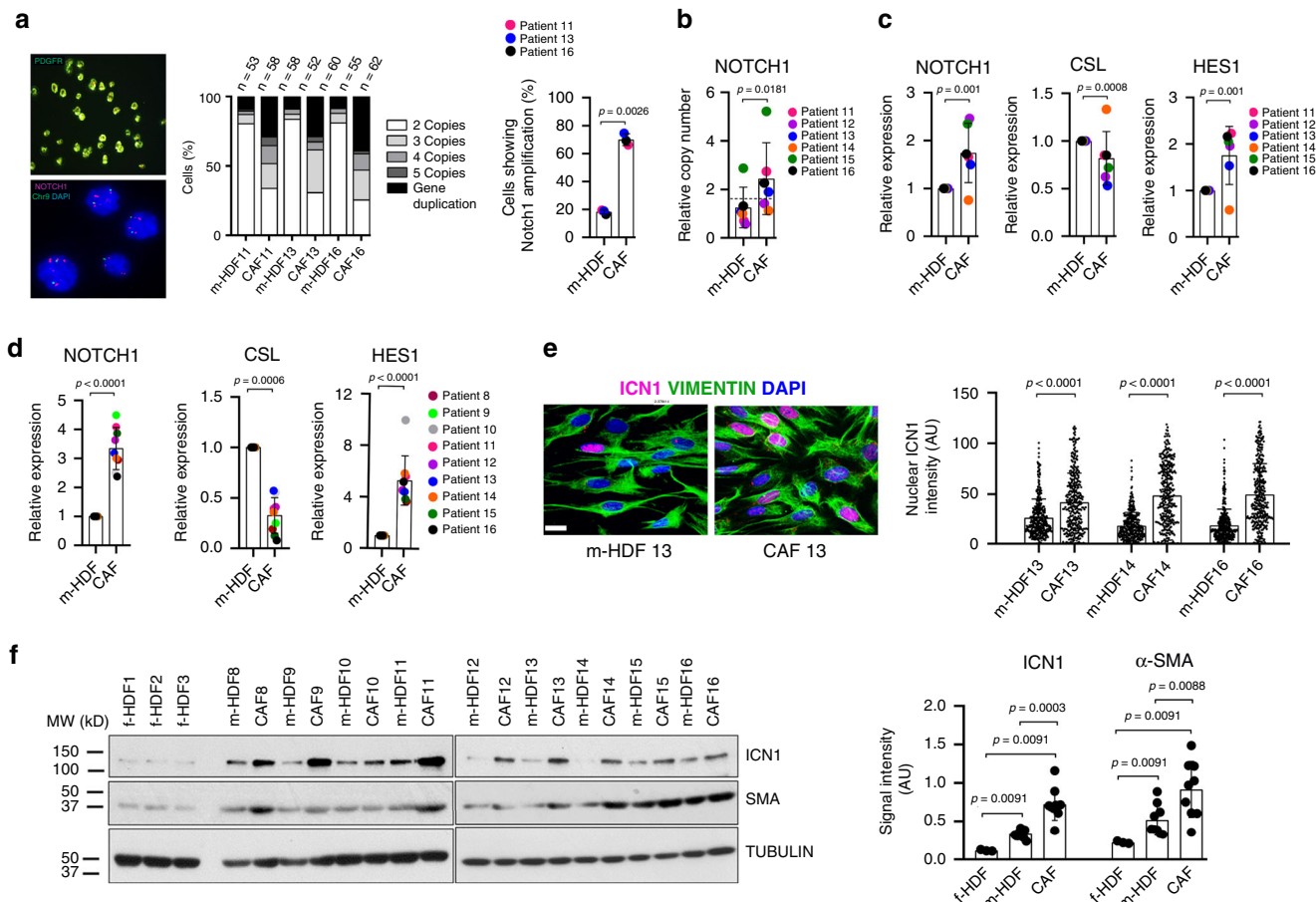

**Fig. 2 *NOTCH1* gene amplification and overexpression in CAFs. a** Fluorescence-guided laser capture micro-dissection (LCM) followed by FISH analysis of *NOTCH1* gene copy number in fibroblasts (PDGFRα positive) of SCC lesions versus unaffected flanking skin from the same patients. Representative images and quantification of percentage of cells with *NOTCH1* copy number variations. Frozen blocks of SCC samples were the same from which CAFs analyzed in Fig. 1d were derived. The number of analyzed nuclei (n) obtained from two independent experiments are shown on top of the corresponding bar. Overall quantification and statistical significance of percentage of cells with *NOTCH1* amplification in CAFs versus m-HDFs from three different patients. Mean ± s. d. Two-tailed paired *t*-test, \*\**p* < 0.01. *n*(patients, CAF) = 3, *n*(patients, m-HDF) = 3. Additional representative images are shown in Supplementary Fig. 1a. **b** LCM followed by qPCR analysis of *NOTCH1* gene copy number in fibroblasts (PDGFRα positive) from SCC lesions (CAF) and unaffected flanking skin (m-HDF) of same patients as in Fig. 1d, compared to similarly captured cells from three unmatched healthy donors. Mean ± s.d. Two-tailed paired *t*-test, \**p* < 0.05. *n*(patients, CAF) = 6, *n*(patients, m-HDF) = 6, and *n*(healthy donor, f-HDF) = 3. **c** LCM followed by RT-qPCR analysis of *NOTCH1*, *CSL*, and *HES1* mRNA expression in laser-captured fibroblasts associated with SCC (CAF) versus fibroblasts from flanking unaffected skin (m-HDFs) from the same samples as in **b**. Mean ± s.d. One sample *t*-test, \*\*\**p* < 0.001. *n*(patients, CAF) = 6, *n*(patients, m-HDF) = 6. **d** RT-qPCR analysis of *NOTCH1*, *CSL*, and *HES1* mRNA expression in early passage SCC-derived fibroblasts versus matched fibroblasts from flanking unaffected skin from the same samples as in Fig. 1d. Mean ± s.d. One sample *t*-test, \*\*\**p* < 0.001, \*\*\*\**p* < 0.0001. *n*(CAF strain) = 9, *n*(m-HDF strain) = 9. **e** Representative images and quantification of immunofluorescence analysis of ICN1 (magenta) coupled with VIMENTIN (green) of m-HDF and CAF strains. ICN1 signal intensity for each individual cell is indicated by scatter dot plot. Scale bar, 10 μm. >343 cells were counted per sample. Mean ± s.d, two-tailed Mann–Whitney test, \*\*\*\**p* < 0.0001. *n*(CAF strain) = 3, *n*(m-HDF strain) = 3. **f** Immunoblotting with antibodies against ICN1, α-SMA, and γ-TUBULIN of f-HDFs, m-HDFs, and CAFs as in Fig. 1d. Densitometric quantification of ICN1 protein levels after γ-TUBULIN normalization. Mean ± s.d, two-tailed Mann–Whitney test, \*\**p* < 0.0, \*\*\**p* < 0.001. *n*(CAF strain) = 9, *n*(m-HDF strain) = 9, and *n*(f-HDF strain) = 3.

a dose response of γ-H2AX induction, as a marker of DDR, in three patient-derived m-HDF strains after repeated UVA treatments (Supplementary Fig. 3a). FISH analysis of the treated cultures showed that the fraction of m-HDF cells harboring *NOTCH1* gene amplifications doubled upon repeated UVA exposure (Fig. 4a). Results were validated by qPCR analysis, showing a dose-dependent increase of *NOTCH1* gene copy number in the UVA-treated versus control cultures (Fig. 4b). A similar increase of *NOTCH1* gene copy number was found in independent experiments with the same m-HDF strains upon chronic UVA treatment (Fig. 4c and Supplementary Fig. 3b). Importantly, similar treatment of multiple foreskin-derived f-HDF strains resulted in no increased in

*NOTCH1* gene copy number, indicating that the increase of *NOTCH1* copies is not a direct consequence of UVA treatment (Fig. 4c).

The increased percentage of m-HDFs with *NOTCH1* gene amplification in UVA-treated cultures was paralleled by an increased fraction of cells with nuclear ICN1 expression, while no such increase was found with similarly treated f-HDFs (Fig. 4d). In fact, UVA treatment of these cells at the doses used for these experiments was not sufficient to induce expression of the *NOTCH1* or *JAGGED 1* and *2* genes, while causing down-modulation of *CSL* (Supplementary Fig. 3c).

Thus, elevated NOTCH1 levels and activity are required for sustained expression of CAF effector genes and CAF proliferation;

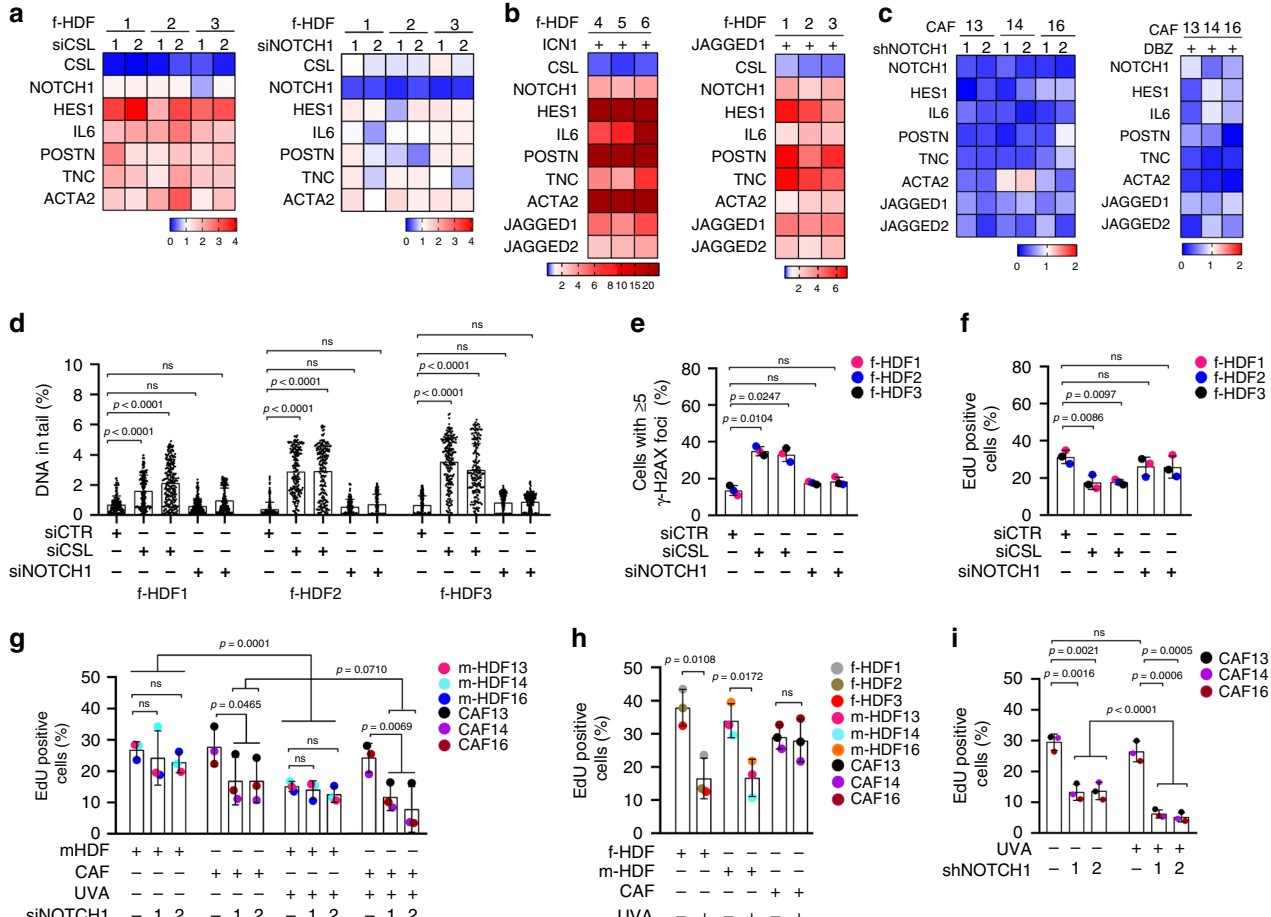

**Fig. 3 Consequences of modulation of *NOTCH1* expression and activity in HDFs and CAFs. a–c** Heatmap of changes in gene expression of the indicated genes in: **a** f-HDF strains plus/minus *CSL* or *NOTCH1* silencing by two different siRNAs versus siRNA controls for 3 days; **b** f-HDF strains infected with a lentivirus for doxycycline-inducible expression of ICN1 versus empty vector control, 5 days after ICN1 induction or cultured on JAGGED 1-coated versus IgG coated control dishes for 3 days; **c** CAF strains infected with two different *NOTCH1* silencing lentiviruses versus empty vector control for 5 days or treated with the γ-Secretase Inhibitor DBZ (10 μM) versus DMSO vehicle alone for 5 days. Results are shown as gene expression folds (down- or up-regulation in blue and red, respectively). Individual bar plots and statistical quantification of the results are shown in Supplementary Fig. 2a–c. **d** Comet assays of f-HDF strains with or without *CSL* and *NOTCH1* silencing. The percentage DNA in tail for individual cells is indicated by scatter dot plot. >184 cells were analyzed per sample. One-way ANOVA followed by Dunnett's test. n(f-HDF strain) = 3. **e** Immunofluorescence analysis with antibodies against γ-H2AX of f-HDF strains plus/minus *CSL* and *NOTCH1* silencing. >201 cells were analyzed per sample. One-way ANOVA followed by Dunnett's test. n(f-HDF strain) = 3. n(CAF strain) = 3. **f–i** EdU assays of: **f** f-HDF strains plus/minus *CSL* and *NOTCH1* silencing. n(cells) >201, One-way ANOVA followed by Dunnett's test n(f-HDF strain) = 3; **g** m-HDF and CAF strains plus/minus siRNA mediated *NOTCH1* gene silencing and plus/minus UVA irradiation. 24 h after siRNA transfection cells were mock treated or irradiated with UVA (500 mJ/cm²), followed by EdU labeling 48 h later, n(cells)> 99, two-way ANOVA between groups and two-tailed unpaired t-test within each group, n(m-HDF strain) = 3, n(CAF strain) = 3; **h** f-HDF, m-HDF, and CAF strains 72 h after UVA irradiation (500 mJ/cm²) versus mock treated, n(cells)>89, two-tailed unpaired t-test, n(CAF strain) = 3. n(f-HDF strain) = 3, n(m-HDF strain) = 3, and n(CAF strain) = 3; **i** CAF strains plus/minus shRNA-mediated *NOTCH1* silencing for 7 days and plus/minus UVA irradiation as in **h**. n(cells) >103, two-tailed unpaired t-test, n(CAF strain) = 3. All data represented as Mean ± s.d., *p < 0.05, **p < 0.01, ***p < 0.001, ****p < 0.0001.

*NOTCH1* gene amplification together with other factors not present in normal foreskin-derived HDFs can contribute to CAFs response to chronic UVA exposure.

**NOTCH1 up-regulation in CAFs blocks the DDR/ATM signaling cascade**. As mentioned above, separately from its CSL-mediated role in transcription, the activated intracellular domain of NOTCH1 (ICN1) was previously reported to bind to ATM in cancer cells and to prevent the ATM/P53 phosphorylation cascade[25]. More specifically, by competitive binding, ICN1 was previously shown to prevent ATM association with FOXO3A, thereby impairing ATM phosphorylation of substrates downstream of γ-H2AX[26]. Such a mechanism could also apply to CAFs and account for their sustained proliferation in spite of their

persistent genomic instability under basal conditions and upon UVA-induced DNA damage.

Proximity ligation assays (PLAs) showed low association of NOTCH1-ATM in foreskin and matched HDFs, while multiple complexes were detectable in CAFs already under basal conditions (Fig. 5a). NOTCH1 binding to ATM was substantially reduced and ATM-FOXO3A association was increased in CAFs by treatment with DBZ (Fig. 5b, c), which suppresses NOTCH1 proteolytic cleavage and activation[27] (Supplementary Fig. 4b). ATM-FOXO3A association was induced by UVA treatment of m-HDFs irrespectively of whether or not *NOTCH1* was silenced (Fig. 5d). By contrast, ATM-FOXO3A complexes were not induced in CAFs by UVA exposure unless the treatment was combined with *NOTCH1* silencing, which was by itself sufficient to induce these complexes in CAFs already under basal

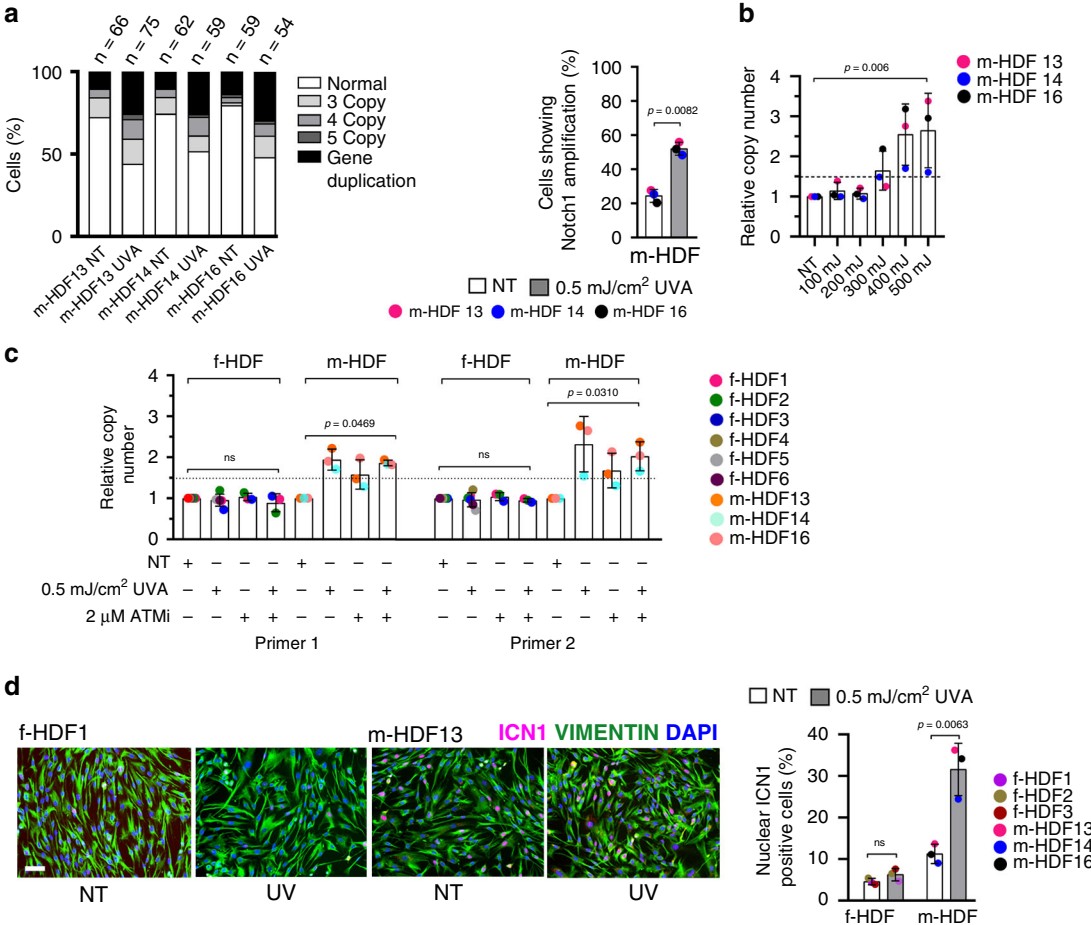

**Fig. 4 Expansion of cultured dermal fibroblast populations with *NOTCH1* gene amplification and protein overexpression upon repeated UVA exposure.** **a** Left panel: quantification of *NOTCH1* gene copy number by FISH with a *NOTCH1*-specific probe in parallel with a probe for *chr9* q21.3 localization in m-HDF strains irradiated with UVA 500 mJ/cm² every other day for three times versus mock-treated control (NT). FISH analysis was performed 72 h after the last exposure. The number of analyzed nuclei (n) was obtained from two independent experiments. Right panel: quantification of the percentage of cells showing *NOTCH1* amplification in m-HDFs plus/minus UVA treatment. Values for each strain are indicated as dots with mean ± s.d. two-tailed unpaired *t*-test, **$p < 0.001$. $n$(m-HDF strain) = 3. **b** Quantification of *NOTCH1* gene copy number in m-HDF strains plus/minus repeated UVA treatment (100, 200, 300, 400, and 500 mJ/cm²) versus mock treated control as in **a**. Primers for *NOTCH1* gene were used together with primers specific for the housekeeping gene *GAPDH* for internal normalization. Values for each strain are indicated as dots with mean ± s.d. One-way ANOVA with Friedman test, **$p < 0.01$. $n$(m-HDF strain) = 3. An additional independent experiment is shown in Supplementary Fig. 3b. **c** Quantification of *NOTCH1* gene copy number in f-HDF versus m-HDF strains plus/minus chronic UVA exposure as in **a**. Parallel cultures were also treated with ATM inhibitor (ATMi) KU-60019 (2 μM) as indicated. Two primers for *NOTCH1* gene were used. Relative DNA copies were calculated after *GAPDH* internal normalization relative to mock-treated control. Values for each strain are indicated as dots with mean ± s.d. One-way ANOVA, *$p < 0.05$. $n$(f-HDF strain) = 6, $n$(m-HDF) = 3. **d** Representative images and quantification of immunofluorescence analysis of ICN1 expression in f-HDF and m-HDF strains upon chronic UVA treatment as in **a** versus mock-treated control (NT). Scale bar, 10 μm. >201 cells were scored per sample. Values for each strain are indicated as dots with mean ± s.d. two-tailed unpaired *t*-test, *$p < 0.05$. $n$(f-HDF strain) = 3, $n$(m-HDF strain) = 3.

conditions (Fig. 5d). Conversely, ICN1 expression in multiple foreskin-derived HDF strains blocked UVA induction of ATM-FOXO3A complexes (Fig. 5e).

These results were verified by co-immunoprecipitation assays showing the exclusive association of ATM with ICN1 and not FOXO3A in multiple CAF strains; strong ATM-FOXO3A association was found in these cells upon suppression of NOTCH1 activation by the γ-secretase inhibitor DBZ (Fig. 5f). Consistent with the above results, immunofluorescence and immunoblot analysis of multiple CAF strains showed that γ-H2AX levels were not significantly affected by *NOTCH1* silencing, while phosphorylation levels of ATM, CHK2, p53, and other downstream ATM substrates (as detected by anti-pS/TQ antibodies), were all strongly induced by *NOTCH1* gene silencing or γ-secretase inhibitor treatment (Fig. 6a–d and

Supplementary Fig. 4a, b). Phosphorylation levels of ATM, CHK2, p53, and γ-H2AX were only induced in HDFs by UVA treatment but not *NOTCH1* gene silencing (Supplementary Fig. 4c). On the other hand, ICN1 expression in these cells was by itself sufficient to suppress phosphorylation of all these proteins in response to UVA exposure (Fig. 6e).

The findings are of functional significance, as proliferation and clonogenicity of CAFs was markedly suppressed by *NOTCH1* knockdown, with no growth suppressing effects occurring in the same CAF strains with CRISPR-mediated TP53 gene deletion, which impairs the ATM/p53 pathway (Fig. 7a, b and Supplementary Fig. 5a, b). Treatment with DBZ and two other γ-secretase inhibitors (DAPT and RO4929097) mirrored the growth inhibitory effects of *NOTCH1* silencing in various CAF strains (Fig. 7c, d). The effects were again specific, since treatment with

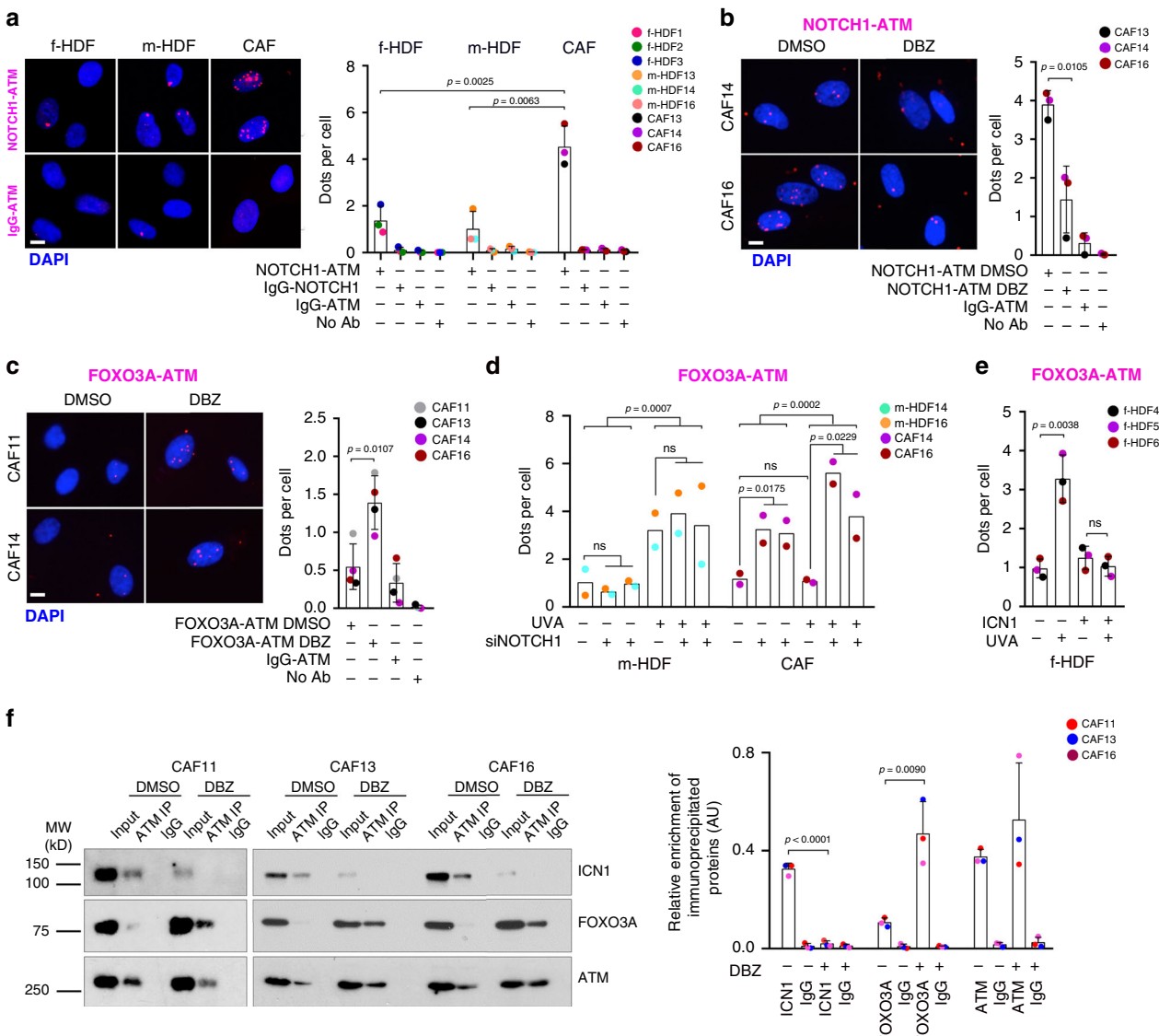

**Fig. 5 NOTCH1-ATM and ATM-FOXO3A complex formation in HDFs and CAFs. a** Proximity ligation assays (PLAs) with antibodies against ATM and NOTCH1 of multiple f-HDF, m-HDF, and CAF strains. **b, c** PLAs of CAF strains plus/minus treatment with DBZ (10 μM) for 5 days with antibodies against ATM and NOTCH1 (**b**) or ATM and FOXO3A (**c**). **d, e** PLAs with antibodies against ATM and FOXO3A of m-HDF and CAF strains plus/minus siRNA-mediated silencing of *NOTCH1* and UVA irradiation as indicated (500 mJ/cm2) (**d**), and of f-HDF strains plus/minus infection with an ICN1 expressing lentivirus for empty vector control for 5 days and UVA irradiation (500 mJ/cm2) for the last 72 h (**e**). IgG-NOTCH1, IgG-ATM, and no primary antibodies were used as negative controls. Shown are representative images and quantification of average number of PLA puncta (dots) per cell per strain, counting in each case. Scale bar, 2 μm. **a** n(cells) >78, n(f-HDF strain) = 3, n(m-HDF strain) = 3, n(CAF strain) = 3; **b** n(cells) > 100, and n(CAF strain) = 3; **c** n(cells) > 74, n(CAF strain) = 4; **d** n(cells) > 74, n(m-HDF strain) = 2, n(CAF strain) = 2; **e** n(cells) > 72, and n(f-HDF strain) = 3. Two-tailed unpaired t-test within groups in **a–e** and two-way ANOVA among groups in **d**. *p < 0.05, **p < 0.01, ***p < 0.001. **f** Co-immunoprecipitation (co-IP) assays of CAF strains plus/minus DBZ (10 μM) treatment for 5 days with anti-ATM antibodies or non-immune IgG followed by immunoblotting, together with equal input amounts, with antibodies against the ICN1, FOXO3A, and ATM proteins. Right panel: densitometric quantification for the indicated proteins normalized to input. Mean ± s.d. two-tailed unpaired t-test, **p < 0.01, ****p < 0.0001. n(CAF strain) = 3.

DBZ caused no growth suppression of CAFs with *TP53* deletion (Fig. 7c). As predicted from all our other findings, proliferation and clonogenicity of HDFs was unaffected by either *NOTCH1* silencing or inhibition (Fig. 7b, c). The NOTCH inhibitor DBZ was also without effects on proliferation of skin squamous carcinoma cells (SCC13; Fig. 7e), as expected by the growth and tumor suppressing function that NOTCH activation plays in these cells[28]. Supporting the ATM-dependency of the effects, growth inhibition of CAFs by *NOTCH1* silencing was also rescued by treatment with a pharmacological ATM inhibitor at lower doses than those used to suppress growth of cancer

cells[29,30], which by themselves caused little or no effects on proliferation of these cells (Fig. 7f and Supplementary Fig. 5c).

Thus, suppression of *NOTCH1* expression or activity in CAFs unleashes the DDR/ATM signaling cascade and TP53-dependent growth suppression.

**Loss of NOTCH1 expression impairs cancer/stromal cell expansion.** A defining property of CAFs is to promote proliferation of neighboring cancer cells[1,5]. We recently developed an in vitro cancer/stromal cell expansion assay based on the co-culture in matrigel of fluorescently labeled SCC cells and CAFs[23].

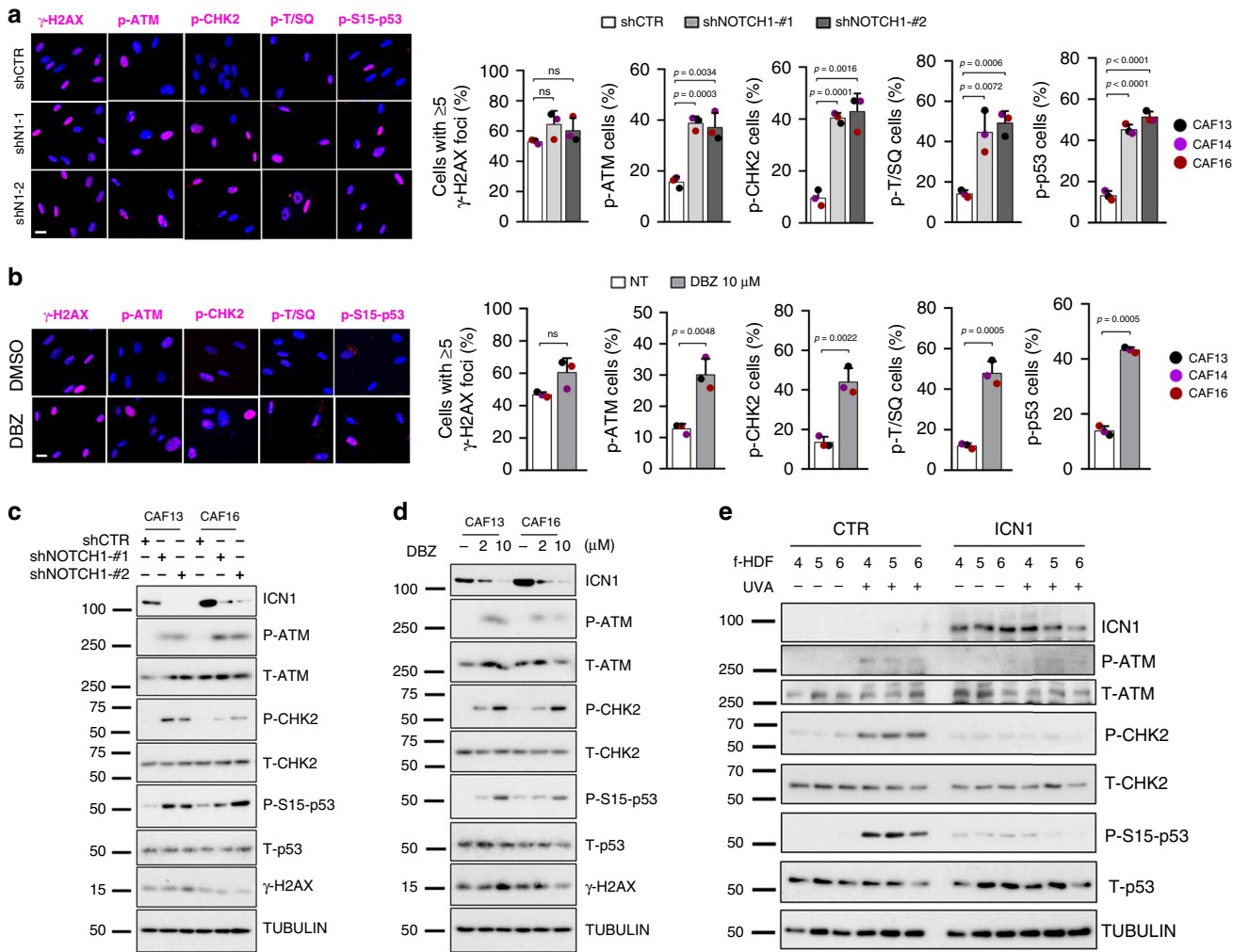

**Fig. 6 Increased NOTCH1 expression in CAFs suppresses ATM/P53 signaling. a, b** Representative images and quantification of immunofluorescence analysis with antibodies against the indicated proteins/epitopes of CAF strains plus/minus shRNA-mediated *NOTCH1* silencing (**a**) or plus/minus DBZ treatment (10 μM) for 5 days (**b**). Scale bar, 10 μm. pS/TQ refers to the antibody-recognized phosphorylated SQ/TQ cluster domains that are structural hallmarks of various ATM and ATR substrates[40]. Values for each strain are indicated as dots with mean ± s.d. **a** $n$(cells) > 194 per sample. One-way ANOVA followed by Dunnett's test; **b** $n$(cells) > 167 per sample. Two-tailed unpaired $t$-test. **$p < 0.01$, ***$p < 0.001$, ****$p < 0.0001$. $n$(CAF strain) = 3. Immunoblot analysis of ICN1 expression levels is shown in Supplementary Fig. 4a, b. **c, d** Immunoblot analysis of two CAF strains plus/minus shRNA-mediated *NOTCH1* silencing (**c**) or plus/minus DBZ treatment (2 and 10 μM) for 5 days (**d**), with antibodies against P-ATM, P-CHK2, P-S15-p53, and respective total proteins, ICN1, γ-H2AX, and γ−TUBULIN (as equal loading control). The same blots were stripped and re-probed. **e** Immunoblot analysis of P-ATM, P-CHK2, P-S15-p53, and respective total proteins and γ-TUBULIN (as equal loading control) of f-HDF strains infected with an ICN1 expressing lentivirus versus empty vector for 5 days plus/minus UVA irradiation (500 mJ/cm²) or mock treatment for the last 72 h. The same blots were stripped and re-probed. $n$(f-HDF strain) = 3.

As shown in Fig. 8, the formation of large clusters of SCC13 cells was severely reduced in the presence of multiple CAF strains with *NOTCH1* gene silencing, with similar suppressive effects being elicited by treatment of these cultures with the γ-secretase inhibitor DBZ, which, as shown above (Fig. 7e), causes no direct growth inhibition of SCC cells.

To validate the above findings in vivo, we resorted to an orthotopic skin cancer model, based on mouse ear injections of combinations of cells plus/minus various genetic manipulations[3,24]. Parallel injections were performed with DsRed-expressing SCC cells (SCC13) admixed with two different CAF strains (CAF14 and CAF16) plus/minus *NOTCH1* silencing. In vivo imaging over a 2 weeks period showed reduced time-dependent expansion of the fluorescently labeled SCC13 cells in the presence of CAFs with silenced *NOTCH1* versus controls and a smaller tumor volume at the time of collection (Fig. 9a). Immunofluorescence analysis at the end of the experiment

showed significantly lower Ki67 positivity of CAFs with *NOTCH1* silencing and of associated SCC cells relative to both types of cells in control lesions (Fig. 9b and Supplementary Fig. 6a). CAFs were identified by the use of human-specific anti-VIMENTIN antibodies with parallel staining with pan-Vimentin antibodies detecting these cells and also mouse fibroblasts further away from the tumor area (Supplementary Fig. 6c). Levels of the CAF marker PERIOSTIN (POSTN) as well as markers of macrophage infiltration (CD68) and angiogenesis (CD31) were also significantly reduced in lesions formed by CAFs with silenced *NOTCH1* (Fig. 9c and Supplementary Fig. 6b). CAFs in these lesions were characterized by strong positivity for the phosphorylated forms of ATM, CHK2, p53, and other ATM substrates, paralleling the in vitro findings (Fig. 9d).

The similar effects of DBZ treatment and *NOTCH1* silencing on cultured CAFs and on SCC/CAF co-cultures suggested that this compound could also be beneficial in vivo, for suppression of

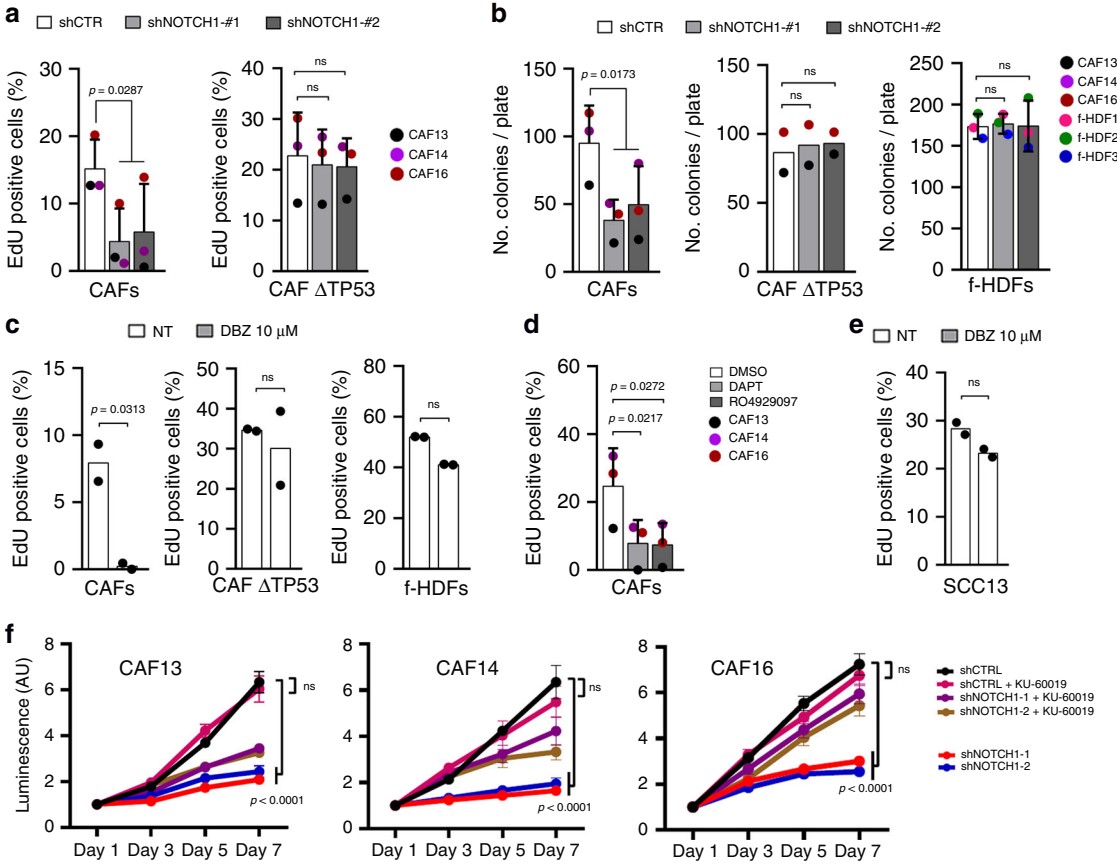

**Fig. 7 NOTCH1 silencing or inhibition impair CAFs proliferation. a** EdU assays of CAF strains plus/minus *NOTCH1* silencing (7 days) and of the same CAF strains with CRISPR-mediated disruption of the *TP53* gene (as characterized in Supplementary Fig. 5a, b) plus/minus *NOTCH1* silencing (7 days). >196 cells were analyzed per condition. n(CAF strain) = 3, n(CAF CRISPR TP53 strain) = 3. **b** Clonogenicity assays of CAF strains plus/minus shRNA-mediated *NOTCH1* silencing (10 days), of CAF strains with CRISPR-mediated disruption of the TP53 gene plus/minus *NOTCH1* silencing (10 days), and of three f-HDF strains plus/minus shRNA-mediated *NOTCH1* silencing (10 days). n(technical replicates/condition) = 3, n(CAF strain) = 3, n(f-HDF strain) = 3, and n(CAF CRISPR TP53 strain) = 2. **c** EdU assays of CAF strains (#13 and 16) plus/minus CRISPR-mediated disruption of *TP53* and of f-HDF strains (#5 and 6) after 5 days of treatment with DBZ (10 μM) versus DMSO vehicle alone. >125 cells were analyzed per condition. n(CAF strain) = 2, n(CAF CRISPR TP53 strain) = 2, and n(f-HDF strain) = 2. **d** EdU assays of CAF strains plus/minus treatment with the γ-secretase inhibitors DAPT and RO4929097 (10 μM) versus DMSO vehicle alone for 5 days. >143 cells were analyzed per condition. n(CAF strain) = 3. **e** EdU assays of SCC13 strain after 5 days of treatment with DBZ (10 μM) versus DMSO vehicle alone. >317 cells were analyzed per condition. n(SCC13) = 1. **a–e** Values for each strain are indicated as dots with mean ± s.d. two-tailed unpaired t-test, *p < 0.05. **f** CellTiter-Glo analysis of CAF strains infected with two *NOTCH1* silencing lentiviruses versus empty vector control (for 7 days) plus/minus concomitant treatment with the ATM inhibitor KU-60019 (2 μM) at the indicated days after viral infection. Results are presented as luminescence intensity values relative to day 1. Data are shown as mean of values in triplicate dishes ±sd. Two-way ANOVA followed by Tukey's multiple comparisons test, ****p < 0.0001. n(CAF strain) = 3.

cancer/stromal cell expansion. This possibility was assessed by the same assays described above, by mouse ear injection of fluorescently labeled SCC13 cells admixed with CAFs, followed, 24 h after injection, by topical treatment with DBZ in parallel with ethanol vehicle alone. As observed upon *NOTCH1* silencing, in vivo DBZ-treatment reduced expansion of lesions over time, decreasing both SCC cells and CAFs proliferation (Fig. 10a, b). Levels of the POSTN, CD68, and CD31 markers were also significantly reduced in lesions treated with the DBZ compound (Fig. 10c), while phosphorylation of ATM, CHK2, p53, and other ATM substrates was strongly increased (Fig. 10d).

Thus, in an orthotopic skin cancer model, genetic or pharmacologic inhibition of NOTCH signaling in CAFs is sufficient to suppress CAF effectors gene expression and impair cancer and stromal cell expansion.

## Discussion

The NOTCH/CSL signaling pathway plays a key role in early steps of CAF activation. Loss of CSL transcription repressive

function leads to concomitant induction of stromal fibroblast senescence and up-regulation of a large battery of CAF-effector genes[3,23,24]. This can be the combined result of down-modulation of CSL expression by exogenous pro-carcinogenic insults, such as UVA or smoke exposure[17,31], and of NOTCH activation, which converts CSL from a repressor into an activator of transcription[3]. While converging on gene expression, both NOTCH and CSL can play additional independent functions[15,32]. Separately from NOTCH, we recently found that CSL can bind directly to telomeric DNA and anchor other protective proteins to telomeres thereby ensuring chromosomal integrity in HDFs[19]. On the other hand, we have shown here that, in CAFs with compromised CSL function and increased genomic instability, *NOTCH1* gene amplification and elevated expression inhibit the DDR/ p53 signaling cascade and growth arrest. At the same time, increased NOTCH1 activity in these cells ensures elevated expression of a battery of CAF effector genes with established tumor-promoting functions. Both mechanisms are interrupted by genetic or pharmacological inhibition of NOTCH1 activation in

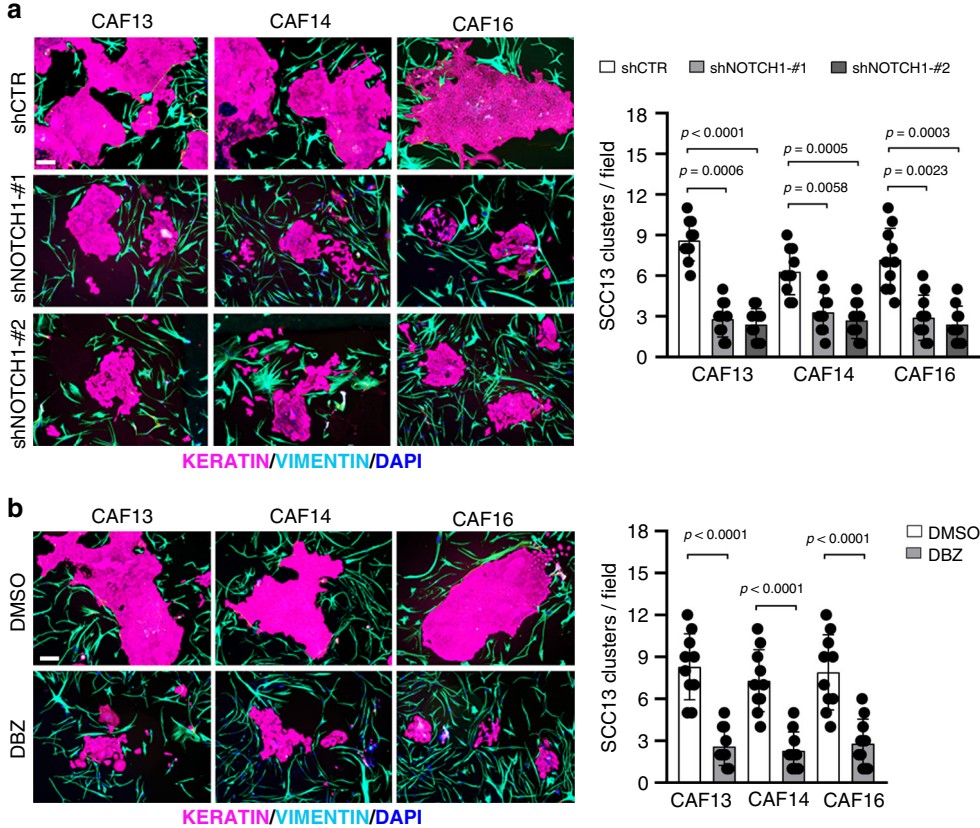

**Fig. 8 *NOTCH1* silencing or inhibition suppress CAFs-mediated proliferation potential of SCC13 cancer cells. a, b** Representative images and relative quantification of immunofluorescence analysis with anti-PANKERATIN (magenta) and anti-VIMENTIN (cyan) antibodies of SCC13 cells co-cultured at a 1:1 ratio with CAF strains infected with a *NOTCH1* silencing lentivirus versus empty vector control for 3 days (**a**) or treated with DBZ (10 μM dose) versus DMSO for 3 days (**b**). Graphs show the quantification of large SCC13 cell clusters (>40 cells) per examined field. Scale bar, 200 μm. Values for each field are indicated as dots with mean ± s.d. **a** One-way ANOVA with Dunnett's test and **b** two-tailed unpaired *t*-test, **$p < 0.01$, *** $p < 0.001$, ****$p < 0.0001$, $n$(CAF strain) = 3. $n$(number of fields, per sample) = 10.

CAFs, thereby accounting for suppression of cancer/stromal cell expansion.

While genetic alterations have widely been explored in the epithelial compartment of tumors, their presence in cancer stromal fibroblasts has been an argument of contention (as reviewed by refs. [5,12,13]). Persistent DNA damage, telomere abrasion, and chromosome alterations occur in dermal fibroblasts with silencing or deletion of the *CSL* gene as well as in skin SCC-derived CAFs, in which CSL is down-modulated[19]. Genomic instability resulting from decreased CSL levels and recurrent UVA exposure, with cumulative effects in human aging populations, could provide a platform for selection of aberrant stromal cells capable of expanding in skin cancer fields. Consistent with this possibility, aCGH analysis revealed multiple CNVs in CAFs at very early passage from SCC dissociation. A high number of aCGH positive probes encompassed the *NOTCH1 gene*, which was found amplified in the majority of cultured CAFs as well as in the corresponding laser-captured fibroblasts of SCC lesions from which CAFs were derived.

While aCGH and qPCR approaches provide an average signal coming from a pool of cells, FISH analysis conveys a more definitive assessment of specific CNVs at the single cell level. Analysis of CAF strains with the latter technique showed heterogeneous cell populations, with a significant fraction of cells harboring *NOTCH1* gene duplication and gains of up to five copies. *NOTCH1* copy number variations, mostly duplications, were also found in a minority of HDFs from apparently unaffected skin surrounding the SCC lesions. Importantly, direct FISH analysis of laser-captured CAFs and matched HDFs, in the absence of possible artifacts introduced by culturing conditions, confirmed the presence of genetic alterations in fibroblast populations that extend beyond areas of overt tumor formation in cancer fields.

We found an overall agreement between *NOTCH1* gene amplification in CAFs and increased expression relative to matched and foreskin HDFs. Various levels of *NOTCH1* expression in CAFs are likely due to several possible mechanisms besides increased copy number, including different duplication endpoints, with differential impact on distinct regulatory elements, and other co-determining factors, including genomic alterations at other loci.

The findings are of likely functional significance, as CAFs with *NOTCH1* gene amplification and increased expression were resistant to UVA-induced growth arrest, and m-HDFs from flanking skin with *NOTCH1* gene amplification were selectively expanded upon repeated UVA exposure.

Previous publications showed that the proteolytically cleaved form of the NOTCH1 receptor can suppress the DNA damage response downstream of γ-H2AX phosphorylation in multiple cancer cell lines[25,26]. Mechanistically, the ICN1 protein was shown to bind to ATM and block its interaction with FOXO3A, thus impairing anchoring and phosphorylation of downstream components of the ATM cascade[26]. Our findings indicate that this mechanism of action takes place also in CAFs, which display

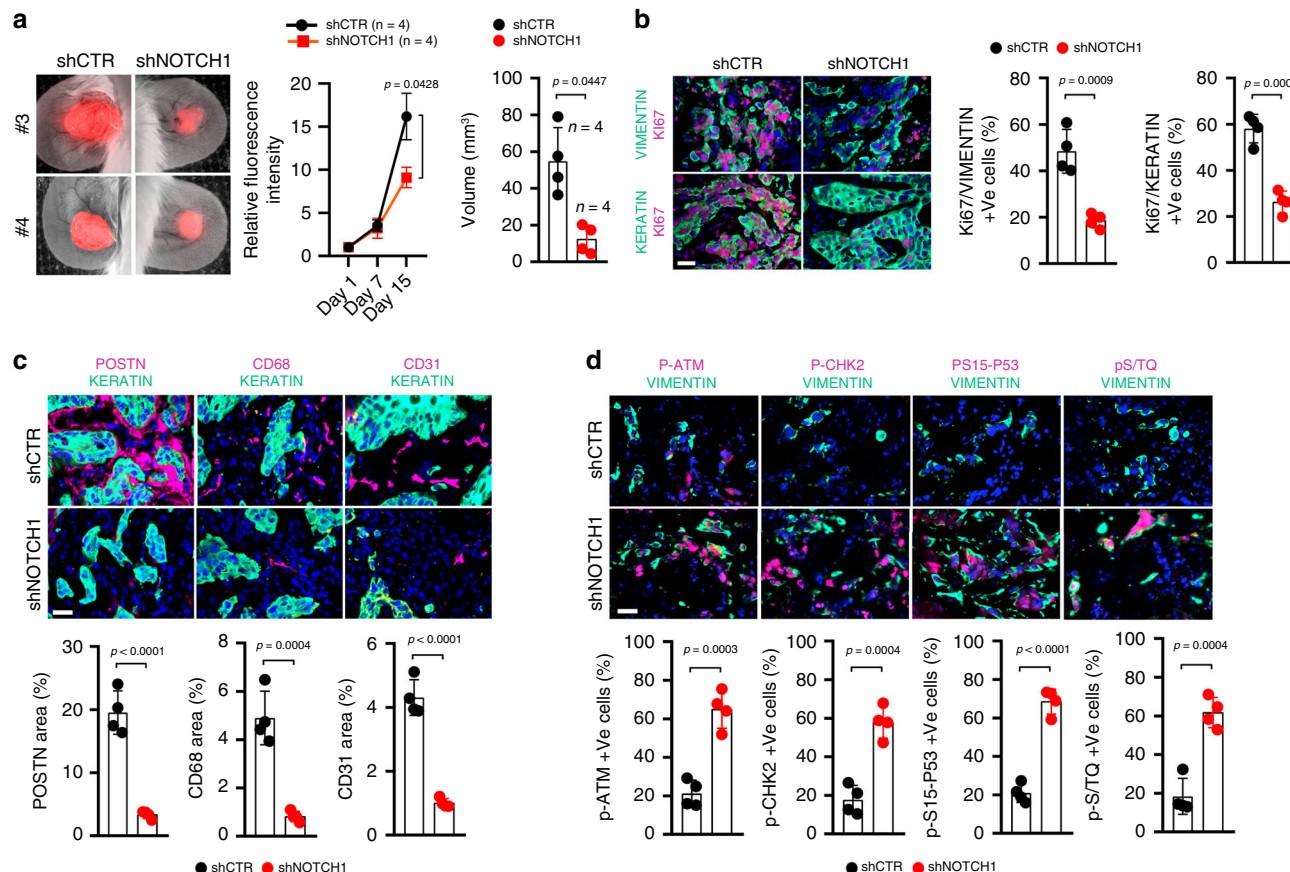

**Fig. 9 *NOTCH1* silencing impairs cancer/stromal cell expansion in vivo. a** DsRed2-expressing SCC13 cells were admixed with CAFs (CAF14) plus/minus *NOTCH1* silencing followed by parallel ear injections into NOD/SCID mice. Shown are images of two ear pairs 15 days after injection, quantification of red fluorescence signal (normalized to day 1 after injection) corresponding to SCC13 cell expansion, and tumor volumes ($V = $ (length × width$^2$) × 0.5). Statistical analysis of fluorescence intensity values over time course of the experiment was calculated by Two-way ANOVA, *$p < 0.05$. Tumor volume values are indicated as dots with mean ± s.d. two-tailed unpaired *t*-test, *$p < 0.05$. $n$(shCTR ear lesion) = 4, $n$(shNOTCH1 ear lesion) = 4. **b** Immunofluorescence analysis of ear lesions with antibodies against Ki67 (magenta) and human-specific VIMENTIN or KERATIN (cyan) for cell identification. Scale bar, 50 μm. Values for each ear lesion are indicated as dots with mean ± s.d. two-tailed unpaired *t*-test, ***$p < 0.001$. $n$(mice) = 4, $n$(shCTR ear lesion) = 4, $n$(shNOTCH1 ear lesion) = 4. Results from an independent experiment with a second CAF strain (CAF16) are in Supplementary Fig. 6a. **c** Immunofluorescence analysis of ear lesions with antibodies against the CAF marker PERIOSTIN (POSTN), macrophage (CD68), and endothelial (CD31) markers (magenta) with KERATIN (cyan). Scale bar, 50 μm. Values for each ear lesion are indicated as dots with mean ± s.d. two-tailed unpaired *t*-test, ***$p < 0.001$, ****$p < 0.0001$. $n$(shCTR ear lesion) = 4, $n$(shNOTCH1 ear lesion) = 4. Results from an independent experiment with a second CAF strain (CAF16) are in Supplementary Fig. 6a. **d** Immunofluorescence analysis of ear lesions with antibodies against the indicated proteins (magenta) together with human-specific anti-VIMENTIN antibodies for CAF identification (cyan). Scale bar, 50 μm. Values for each ear lesion are indicated as dots with mean ± s.d. two-tailed unpaired *t*-test, ***$p < 0.001$, ****$p < 0.0001$. $n$(shCTR ear lesion) = 4, $n$(shNOTCH1 ear lesion) = 4.

elevated levels of DNA damage and ICN1-ATM complex formation. Suppression of NOTCH1 expression or activity in these cells resulted in increased ATM-FOXO3A complex formation, under both basal conditions and upon UVA exposure, and restored the ATM/P53 signaling cascade. Consistent with this mechanism of action, increased ICN1 expression in normal foreskin-derived HDF strains was by itself sufficient to block UVA-induced ATM-FOXO3A complex formation and downstream ATM signaling.

An interesting possibility is that, in parallel with the above, ICN1 binds and sequesters CSL away from telomeres, thereby enhancing the recently demonstrated phenotype of genomic instability[19]. Consistent with this possibility in ongoing work we have found that ICN1 expression in multiple f-HDF strains results in loss of CSL at telomeres and DNA damage. However, this could also be an indirect consequence of down-modulation of CSL expression caused by NOTCH1 activation. Increase genomic instability in f-HDFs with activated NOTCH1 expression was not associated with apoptosis but growth suppression, which can

result from a second independent mechanism, involving induction of *CDKN1A* expression, a direct target of CSL transcriptional repression overcome by NOTCH activation. Because of the complexities involved, further detailed studies will be required to investigate this interesting topic.

While inhibition of the DDR/p53 signaling cascade by increased NOTCH1 activity has the potential of increasing DNA damage, the apoptotic response is also blocked and the identification of bypassing mechanisms triggering this process could be of substantial translational significance. It has also been previously reported that ATM deficiency can be linked to aberrant double-strand DNA repair and chromosomal alterations leading to thymic lymphoma development[33]. Even in the present setting, ATM deficiency may not only be a consequence but to some degree a cause of *NOTCH1* gene amplifications and increased expression. Experimentally, however, treatment of foreskin-derived HDFs with an ATM inhibitor plus/minus prolonged UVA treatment caused no increase in *NOTCH1* gene copy number; this was instead increased in cultures of patients-derived

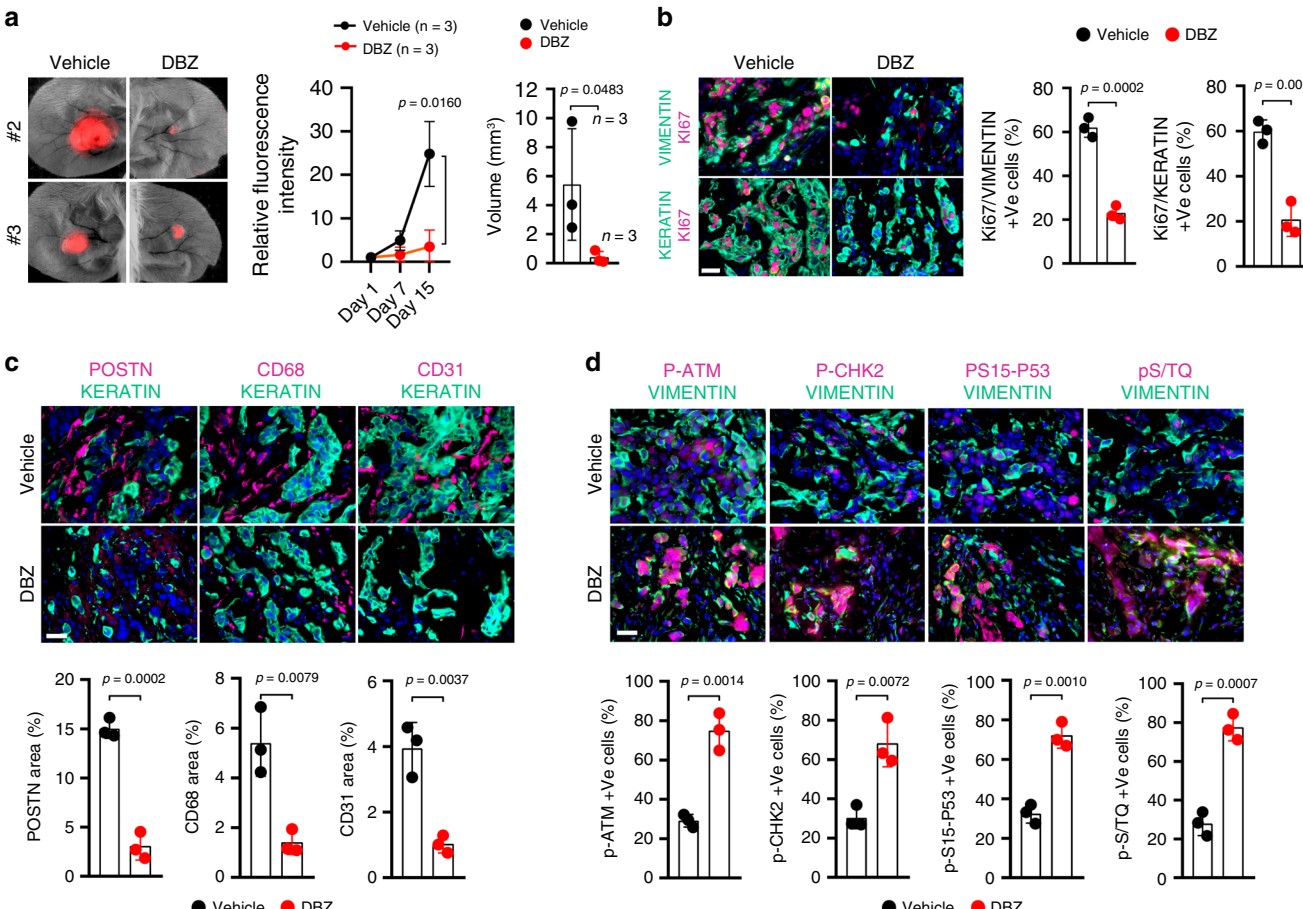

**Fig. 10 Topical treatment with NOTCH1 inhibitors impairs cancer/stromal cell expansion in vivo. a** DsRed2-expressing SCC13 cells admixed with CAFs (CAF13) were injected into ears of NOD/SCID mice, followed by topical treatment with DBZ (1 mM) or Ethanol vehicle alone daily for 1 week starting 24 h after injection. Mice were killed 15 days after injection. Shown are images of two ear pairs 15 days after injection, quantification of red fluorescence signal (normalized to day 1 after injection) corresponding to SCC13 cell expansion and tumor volumes ($V = (\text{length} \times \text{width}^2) \times 0.5$). Statistical analysis of fluorescence intensity values over time course of the experiment was calculated by Two-way ANOVA, *$p < 0.05$. Tumor volume values are indicated as dots with mean ± s.d. two-tailed unpaired $t$-test, *$p < 0.05$. $n$(CTR ear lesion) = 3, $n$(DBZ ear lesion) = 3. **b–d** Representative images and quantification of immunofluorescence analysis of ear lesions with antibodies against the indicated proteins as in Fig. 9b–d. Scale bar, 50 μm. Values for each lesion are indicated as dots with mean ± s.d. two-tailed unpaired $t$-test, **$p < 0.01$, ***$p < 0.001$. $n$(CTR ear lesion) = 3, $n$(DBZ ear lesion) = 3.

m-HDFs, consistent with our other results showing the presence in these cultures of subpopulations of cells with *NOTCH1* gene amplification that are selectively expanded upon prolonged UVA treatment (Fig. 4c).

The findings are of functional significance as *NOTCH1* gene silencing or pharmacological inhibition impaired CAFs proliferation, without any effects on normal HDFs, and suppressed skin SCC/CAF cells expansion. Multiple NOTCH inhibitors are currently being tested for their possible application in cancer treatment. Our findings indicate that these compounds could be effectively used in stroma-targeted cancer prevention and treatment. More specifically, topical treatment with NOTCH inhibitors could be of special interest for organ transplant recipient patients undergoing immune suppressive therapies, for whom skin field cancerization represents an important cause of morbidity and mortality.

## Methods

**Cell manipulations**. HDFs were prepared from discarded foreskin or abdominoplasty skin samples at the Department of Dermatology, Massachusetts General Hospital (Boston, Massachusetts, USA) with institutional approval (2000P002418), or were previously obtained[3]. Conditions for culturing cells, viral shRNA infection, siRNA transfection, qPCR, and RT-qPCR were as in refs. [2,3,34]. Pairs of CAFs and matched HDFs from discarded skin SCC and flanking unaffected areas from the same (anonymized) patients, derived as in Goruppi et al.[34], were given specific

identifiers as indicated in the different panels. CAF strains were used at very early passage of culturing (2nd–3rd passage) unless otherwise specified. A list of cell strains is provided in Supplementary Data 2. For in vivo approaches, skin-derived SCC13 cells[35] were infected with a DsRed2-expressing lentivirus[3]. 2D co-culture assays in CAFs were performed as in Clocchiatti et al.[23].

CAFs and/or SCC13 cells were treated with the following chemicals at the indicated concentrations every 24 h for 5 days: 2 or 10 μM of DBZ (Deshydroxy LY-411575, Sigma), 0.5 or 2 μM of KU-60019 (Sigma), 10 μM of DAPT (γ-secretase inhibitor IX CAS 208255-80-5, Calbiochem), and 10 μM of RO4929097 (Selleckchem) versus vehicle alone. HDFs were treated with a Bio-Link cross-linker UV irradiation system (Vilber Lourmat) equipped with a UVA lamp (375 nm), as indicated in the figure legends. For acute UVA experiments, HDFs were treated the day after seeding with 500 mJ/cm² UVA and samples were collected 72 h after exposure. For chronic UVA experiments, HDFs were treated the day after seeding with incremental doses of UVA (100 mJ/cm², 200 mJ/cm², 300 mJ/cm², 400 mJ/cm², and 500 mJ/cm²). Treatments were performed every other day for three times and samples were collected 72 h after the last exposure. A portable photometer IL1400A (International Light Technologies) was used for dosage determination.

For disruption of the *TP53* gene, CRISPR technology was applied using the same viral vector as in Procopio et al.[3]. Analysis of *TP53* deletion was performed by PCR with a forward primer spanning the deletion site (Supplementary Data 3 and Supplementary Fig. 5a) and by immunoblot in comparison with HDFs upon overnight treatment with 1 μM Doxorubicin versus control (Supplementary Fig. 5b).

HDFs or CAFs were infected with retroviruses as in Procopio et al.[3]. All experiments were carried out with antibiotic resistance selection.

The oligonucleotides used in qPCR and RT-qPCR are provided in Supplementary Data 3 and 5. A detailed list of all the antibodies and the conditions used is in Supplementary Data 4. The siRNA oligonucleotides identifiers for *CSL*

and *NOTCH1* silencing are provided in Supplementary Data 6. CAF strains were stably infected with a lentiviral vector for silencing of *NOTCH1* in parallel with empty vector control, a gift from Dr. T. Kiyono. HDF strains were stably infected with a lentiviral expression vector for inducible expression of activated *NOTCH1* in parallel with empty vector control using the pInducer20 system as in Lefort et al.[36].

Treatment of HDFs with Jagged-1 ligand was performed as in Procopio et al.[3] using Rabbit Anti-Human IgG (Sigma-l2011) plus/minus Jagged-1 (human):Fc (human) (recombinant protein, AG-40A-0081) for coating of the dishes.

**Co-immunoprecipitation, immune detection, and cell assays**. Co-IPs of CAFs treated with DBZ (10 μM for 5 days) versus vehicle were performed as in Procopio et al.[3]. Briefly, cells were lysed in NP-40 buffer and 500μg protein extracts were incubated overnight at 4 °C with 10 μg primary antibodies against the total ATM protein in parallel with corresponding non-immune IgGs. This incubation was followed by the addition of 25 μl of packed Dynabeads Protein A (Invitrogen™, 10002D) and further incubation for 4 h at 4 °C. Beads were washed five times with NP-40 buffer, eluted in 60 μl 2x SDS sample buffer at 95 °C for 15 min, and analyzed by immunoblotting.

Immunofluorescence and immunoblot analyses were performed as in refs. [3,34]. For immunofluorescence, cells were seeded on coverslips, fixed in 4% paraformaldehyde (PFA), and processed as in Goruppi et al.[34]. Immunohistochemistry of tumor and tissue sections was performed as in refs. [2,3,37]. Quantification of γ-H2AX, P-ATM, P-CHK2, pS/TQ, and P-S15-P53 protein levels was made using the watershed algorithm (http://imagej.nih.gov/ij/plugins/watershed.html) and ImageJ 1.8 (NIH). Quantification of all other tissue immunofluorescence staining was performed using ImageJ. Unprocessed original scans of immunoblots are shown in Supplementary Figs. 7–9.

EdU labeling assays were performed using Click-iT EdU Alexa Fluor 488 Imaging kit C10337 (Thermo Fisher Scientific) adding 10 μM EDU (for 3 h in SCC13 and for 5 h in CAFs/HDFs) prior to fixation according to the manufacturer's protocol. Apoptosis assays were performed using Apoptosis/Necrosis Assay Kit ab176749 (Abcam) adding Apopxin Green (for 45 min) according to the manufacturer's protocol. Cell proliferation assays were carried out by measuring the production of ATP using the CellTiter-Glo luminescent assay (Promega) as per the manufacturer's instructions. Images were obtained with a Zeiss Observer Z1 inverted microscope. Clonogenicity assays and alkaline comet assays were performed as in Procopio et al.[3] and Bottoni et al.[19], respectively.

***NOTCH1* gene FISH**. *NOTCH1* gene copy number variation was assessed by FISH with a commercially available probe for the *NOTCH1* locus (spanning a 203-Kb region) combined with one for an independent region of chromosome 9 (9q21.3) following the manufacture's instructions (Empire Genomics, Buffalo, NY). Briefly, cell pellets were resuspended in 0.2% trisodium citrate, 0.2% KCl for 10 min at room temperature. After centrifugation pellets were fixed with freshly made cold Carnoy's fixative and dropped onto glass slides. Slides were heated in denaturation buffer (70% formamide, 2X SSC ph 7.0–8.0) at 73 °C for 5 min and dehydrated sequentially with 70%, 85%, and 100% ethanol for 1 min. In all, 10 μl of *NOTCH1/Chr9* probes mixture (2 μl of probe in 8 μl of hybridization buffer) were applied per slide and incubated at 37 °C for 16 h. Slides were washed with 0.4X SSC, 0.3% NP40 (ph 7.0–7.5) at 73 °C for 2 min followed by one wash with 2X SSC, 0.1% NP40 for 1 min. Slides were counterstained with DAPI and analyzed by Zeiss Axiolmager Z1.

**Proximity ligation assays**. Proximity ligation assays (PLAs)[38] were performed using Duolink PLA kit (Sigma) according to the manufacturer's protocol as in Goruppi et al.[34]. Briefly, samples were fixed in 4% paraformaldehyde (PFA), permeabilized in 0.1% Triton and processed as in Goruppi et al.[34]. Images were obtained with a Nikon Eclipse Ti confocal microscope.

**Array CGH**. For comparative genomic hybridization array (aCGH), total DNA was isolated from CAFs using a DNeasy blood and tissue kit (Qiagen). DNA digestion, labeling, hybridization, and washing were performed following manufacturer's instructions. Briefly, CAF samples were labeled with Cy5 and hybridized to Agilent human genome CGH + SNP Microarrays (Agilent Technologies) together with a Cy3-labeled reference genome (Promega). Agilent SurePrint G3 Cancer CGH + SNP Microarrays contained 180K probes across the whole genome based on the NCBI Build 37. Scanning with an Agilent G2505C Microarray Scanner (Agilent Technologies) was followed by image and data processing using Feature Extraction software version 12 (Agilent Technologies) and Agilent Genomic Workbench version 7 (Agilent Technologies). Aberrant regions were detected using ADM-2 algorithm with threshold set to 6. To avoid false positive calls the minimum number of consecutive probes for amplifications/deletions was set to 3, together with a minimum average absolute Log Ratio ≥ 0.25.

aCGH data are deposited in the public repository (GSE113577).

**LCM experiments**. Skin and SCC frozen samples used for LCM followed by RT-qPCR, qPCR, or FISH were provided by the Department of Dermatology, Massachusetts General Hospital (Boston, Massachusetts, USA), with institutional review boards approvals and informed consent. LCM was performed using an Arcturus XT micro-dissection system (Applied Biosystems) as in refs. [2,3]. Nuclei isolation for *NOTCH1* FISH analysis was performed as in DiFrancesco et al.[39]. The oligonucleotides used in qPCR and RT-qPCR are provided in Supplementary Data 3 and 5.

**Animal studies**. Mouse ear injections of cells were carried out in 8–10-week-old female NOD/SCID/IL2rγ[-/-] mice as in Procopio et al.[3]. DsRed2-expressing SCC13 cells ($1 \times 10^5$) were admixed with equal numbers of CAFs (plus/minus shRNA-mediated *NOTCH1* silencing) and injected (3 μl per injection) using a 33-gauge micro syringe (Hamilton). Mice were kept under standard housing conditions with 12 light/12 dark cycle and temperatures of 65–75 °F (~18–23 °C) with 40–60% humidity. Starting the day after injection, mouse ears were imaged using a fluorescent stereomicroscope (Leica MZ-FLIII), every 7 days for 15 or 21 days. After mice killing, images of the ears were taken using bright field and fluorescence stereomicroscopy. All animal studies were approved by the Massachusetts General Hospital Institutional Animal Care and Use Committee (2004N000170) or were performed according to the Swiss guidelines and regulations for the care and use of laboratory animals, with approved protocol from the Canton de Vaud veterinary office (animal license No. 1854.4e). For in vivo DBZ assays, similar ear injections were performed with DsRed2-expressing SCC13 cells admixed with CAFs. One day after injection, mice were topically treated daily for 1 week with DBZ (10 μg in 20 μl of Ethanol) or vehicle alone, on the right and left ears respectively.

**Statistical analysis**. Data are presented as mean ± SD or ratios among treated and controls, with two to three separate CAF/HDF strains in independent experiments as indicated in the Figure legends. Statistical testing was performed using Prism 8 (GraphPad Software). For genomic analysis and functional testing assays, statistical significance of differences between experimental groups and controls was assessed by one sample *t*-test, two-tailed unpaired or paired *t*-test or 1- or 2-way ANOVA. *P* values < 0.05 were considered as statistically significant. The researchers were not blinded and no strain or result was excluded from the analysis.

**Reporting summary**. Further information on research design is available in the Nature Research Reporting Summary linked to this article.

## Data availability

aCGH data for this study is deposited in GEO with the accession codes GSE113577 (https://www.ncbi.nlm.nih.gov/geo/query/acc.cgi?acc=GSE113577). All other relevant data generated in this manuscript that support the findings of this study are available upon request from the authors. Source data are provided with this paper.

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

## Acknowledgements

We are grateful to Dr. G. Chiorino for the bioinformatic support and Dr. T. Kiyono for *NOTCH1* silencing vectors. This work was supported by grants from the National Institutes of Health (R01AR039190; the content not necessarily representing the official views of NIH), the Swiss National Science Foundation (310030_156191/1), the Swiss Cancer League (KFS-4709-02-2019), and the European Research Council (26075083) to G.P.D.

## Author contributions

A.K., G.B., A.C., S.G., P.B., F.L., and I.G. performed the experiments and/or contributed to analysis of the results. P.O. performed the aCGH bioinformatics analysis. V.N. provided clinical samples. A.K., G.B., and G.P.D. designed the study and wrote the manuscript.

## Competing interests

The authors declare no competing interests.
