## [Peer Review File · Nature Communications]

Reviewers' comments:

Reviewer #1 (Remarks to the Author): This was Reviewer #2 at NCB. Their expertise is in CAFs and the microenvironment

Please find enclosed my review of the manuscript entitled "NOTCH1 gene amplification promotes expansion of cancer associated fibroblast populations in human skin" by Bottoni et al.

In this manuscript, Bottoni et al., report that carcinoma associated fibroblasts isolated from human skin squamous cell carcinoma lesions display genomic aberrations with amplification of the NOTCH1 gene. They propose that NOTCH1 gene amplification, results in increase expression of the NOTCH1 protein, which confers a selective advantage for CAF proliferation and overcomes UVA-dependent DDR-induced growth arrest in CAF. They showed that NOTCH1 and ATM interact specifically in CAF. In a therapeutic perspective, the authors show that suppression of NOTCH1 activity (either by pharmacological inhibition or by genetic ablation) in CAF suppresses CAF expansion in vitro and skin cancer lesion in an orthotopic model of skin SCC. Taken into consideration that this manuscript has been peer reviewed in a recent past, I would like to highlight that it has been greatly improved and I would like to thank the authors for having taken into consideration most of the comments. However, the manuscript has changed considerably since the first submission and that new findings are reported. Therefore, I would rather consider this manuscript has a new submission process.

Altogether, this is an elegant study that brings novelty in the field of tumor and stroma crosstalk in cancer expansion. There is a wide variety of techniques used in this manuscript that are convincing. The use of a large set of human CAF and their apparently normal fibroblasts counterpart is a major strength. The accumulation of genomic aberrations in cancer stromal fibroblasts is still under debates and this manuscript could bring interesting information in this field. However, some supplemental evidences would be required for publication in Nature Communications.

Specific comments

Figure 1 describes, using aCGH, the amplification of the NOTCH1 gene in two lines of CAF (8, 9) while the line 10, no probes for NOTCH1 gene scored. Next, qPCR experiments in 6 others CAF lines further validated the data obtained by aCGH, except for the line 14. Also, this figure describes the presence of up to 5 or more copies of the NOTCH1 gene in all the CAF, compared to their counterparts HDF, in which some NOTCH1 amplification can be found, but no amplification is seen in foreskin fibroblasts (AK1, 2 and 3). However, in Figure 2e, the same three CAF lines used in the aCGH (8, 9 and 10) are the ones that present the fewest difference in NOTCH1 protein expression. It is not clear to me how this discrepancy could be explained. Why is there no error bar in F1b, c and F2b, c, d? The quantification of at least 50 cells to quantify the number of copies is rather low and it is not mentioned how many experimental replicates have been done. To define more specifically the CAF lines presented in this manuscript, an immunoblotting of the aSMA protein in the CAF lines, their HDF counterpart and the foreskin fibroblasts is needed.

Figure 3 aims at demonstrating that NOTCH1 amplification leads to confer a growth advantage for the CAF populations upon UVA exposure. The authors suggest that chronic UVA exposures in HDF cells induce an increase in NOTCH1 gene copies. Here, the comparison of the normal foreskin fibroblasts is missing. Moreover, it has been reported that UVA exposure of normal fibroblasts leads to cell senescence, therefore investigating the induction of NOTCH1 in foreskin fibroblasts in response to UVA seems critical. Also, using Edu staining, the authors conclude that NOTCH1 gene amplification results in a lesser inhibition of DNA synthesis in CAF following UVA exposure. To confirm this finding, the use of siRNA targeting NOTCH1 in both CAF cells and HDF exposed to UVA is required. Is it possible that NOTCH1 amplification protects cells from UVA-dependent induction of apoptosis? Figure 3a, b and c do not present any statistical significance.

This finding also leads to the hypothesis that in skin, accumulation of UVA exposure during life in

non pathological condition could lead to NOTCH1 gene amplification. Investigation of NOTCH1 expression in young vs old skin would reinforce the message.

Figure 4 aims at demonstrating the molecular mechanism of NOTCH1-dependent sustained proliferation following UVa exposure. The demonstration of a binding between NOTCH1 and ATM is relevant and convincing. However, a similar assay performed in foreskin fibroblasts and in HDF counterpart is missing. In this context, is NOTCH1 overexpression in foreskin fibroblasts sufficient to suppress ATM/P53 signalling axis? Similarly, as shown in Figure 5, what would be the level of proliferation in foreskin fibroblasts overexpressing NOTCH1?

What is puzzling to me is, how is it possible to find so much DNA damage foci in cells (gamma-H2AX staining) at basal level? How many foci per nucleus are found? The quantification provided here does not reflect that. The CAF lines used in this article are primary cells with very low passages, it seems unfortunate to me to find so much foci in the nucleus, independently of any damage stimuli.

To finish, the author propose that their findings are of therapeutic importance (Figure 6 and 7). Indeed, using an orthotopic mice model of skin cancer, the author show that inhibition of NOTCH1 (genetic and pharmacologic) in CAF, results in reduce tumor fluorescent intensity. I already made some comments in the original reviewing for this model. I still believe that this is not the best model to study cancer cells and stromal fibroblasts interactions. Indeed, to me, the authors completely exclude a potential role for the host fibroblast. Labelling of human vs mice CAF at the end of the experimentation would give us an information on the relative quantity for each CAF population. This aspect is reinforced when you observe the efficacy of the pharmacological treatment vs the genetic ablation in CAF only. It is impossible to conclude, since the pharmacological treatment will also interact with the cancer cells. Also, the measure of the "relative fluorescent intensity" is not appropriate. Indeed, how could you be sure that 100% of the cancer cells still express de Ds-Red marker? Please, indicate the calculation of the tumor volume instead. The number of mice is not indicated. It is not clear to me what $n(\text{ear paired})=4$ means? Only two mice were used, resulting in only 4 tumors? If yes, the number of mice and of tumors used here must be increased. This is probably the reason why the SD is so low in Figure 6a for shNOTCH1.

Reviewer #2 (Remarks to the Author): This was Reviewer #3 at NCB. Their expertise is in Notch signalling in the skin

This manuscript focuses on a really interesting observation that is the occurrence of Notch1 amplifications (and activation) in dermal CAFs, as a mechanism to prevent the blockage in proliferation that is imposed by the ATM-dependent DDR pathway. Thus Notch1 amplification is selected following UV exposure. To my view, amplification of Notch1 and other genomic regions in the CAF populations is very well demonstrated and seems consistent. However, the possibility that Notch favors CAF proliferation following UV exposure by preventing ATM downstream signaling is not so obvious. Thus, all this part of the work should be extensively substantiated experimentally. I would suggest several experiments although authors can design alternative/additional experimental support. Some examples:

1) WB analysis of CAFs (and control HDF that I interpreted as human dermal fibroblasts) following a UV kinetics looking at P-ATM (p-ATR?) and its downstream elements chk2, H2A.x or 53BP1 would be helpful. In principal, chk2 shouldn't be activated in Notch1 proficient CAFs, and the phenotype reverted following Notch inhibition both genetic and pharmacologic. Equally important, gammaH2Ax levels should be maintained elevated as result of deficient damage repair, which could additionally be measured in comet assays. In addition, and as a general rule, WBs need to include analysis of total levels of proteins that are tested in the phosphorylated form (ATM, p53,

chk2, etc). These controls are particularly relevant in this case since differences shown in p-chk2 are minimal.

2) Levels of Notch1 after sh-RNA treatment or pharmacologic Notch inhibition also need to be shown. In figure 4 the clearest effect of Notch inhibition is the increase in p-S15-p53, which can primarily contribute to proliferation blockage that is shown in figure 5. Authors should include an additional experiment showing that p53 phosphorylation in Notch inhibited cells is dependent on ATM activity.

3) As previously said, additional experiments could help to substantiate the effect of active Notch in DDR signaling, DNA damage repair and proliferation blockage including NHEJ reporter assays, cell cycle profiles of Notch1 silenced CAFs and HDF in addition to ki67 staining in figure 5.

4) In figure 3 how can authors discard the possibility that UV treatment is inducing amplification of Notch1 (and other genes) in non-amplified cells instead of expanding the Notch1 amplified population?

5) Since ATM inhibition is primarily ascribed to active Notch1 IF analysis of Figure 3e requires better images and quantification of IL6+ cells showing ICN1.

6) Could authors test whether Notch is activated in HDF and CAFs following UV treatment, and whether Notch inhibition prevents DDR also in normal fibroblast?

7) To me, it is not evident that ATM pathway should be significantly activated in the absence of external damage (and in fact there are negligible levels of P-ATM in untreated CAFs). Does it mean that CAFs display constitutive damage in comparison with HDF but Notch is preventing upstream ATM activation? In this case, P-ATM activation should be quantitatively measured under basal conditions and following UV exposure. This concern also involves figures 6-7 that address the effects of Notch inhibition in the in vivo cancer model. Although results showing that Notch inhibition reduces tumor growth and CAF proliferation in vivo are really nice, I wonder what is the actual contribution of this Notch/ATM pathway in a model system that is never exposed to UV or external damage. Can authors explain this apparent contradiction?

My general opinion is that this work contains relevant data, but sections linking Notch, UV response, ATM signaling and proliferation need additional experimental support.

A minor concern is the inclusion of a reference as Bottoni et al, submitted.

Reviewer #3 (Remarks to the Author): This reviewer was recruited by us. Their expertise is in DNA damage signalling

In this manuscript Bottoni et al. provide evidence of frequent NOTCH1 gene amplification in Cancer Associated Fibroblast from Squamous Cell Carcinoma, and also of its presence in apparently normal Human Dermal Fibroblasts. Interestingly, this increased NOTCH1 expression provides a proliferative advantage, both to HDFs in response to UV exposure, and to cancer/stromal cells in orthoptic models. Although these are interesting observations that are worth studying deeper, the mechanistic conclusions are not sufficiently substantiated, especially regarding the connection with ATM and the DNA-damage response. With this in mind, and considering the recent publication from the group (which was actually part of the original submission), I do not feel that this manuscript substantially advances our knowledge on how NOTCH1 signaling impacts on CAF biology. Please find some general and specific comments below:

- The statistical analysis of the manuscript needs to be thoroughly revisited. In many cases, experimental replicates are not included, and just based on a single analysis in different cell lines. Furthermore, several strong claims are based on differences for which statistical analysis is lacking or incorrect. For example, I bear serious doubts that a t-test is appropriate for many of the cases in which it is applied.

- Generally, there is some degree of inconsistency regarding the cellular systems used in the different experiments. It is unclear why in some cases matched HDFs and CAFs are used (which is

ideal), but in others, cells are not matched or only one cell type is analyzed. In order to unambiguously establish a cause-effect relationship between NOTCH1 levels and UV resistance/proliferation, authors would need to overexpress NOTCH1 in HDFs and compare their response.

- Regarding the effects on the DDR (Figure 4), it is difficult to understand how more than 50% of the cells are positive for gH2AX in the absence of exogenous DNA damage. Authors claim that this is increased DNA damage, but, compared to what? These outstanding levels of damage cast serious doubts on the physiological relevance of the results. Furthermore, in order to directly test their hypothesis, authors would need to check how NOTCH1 depletion specifically affects ATM-mediated signaling of UV damage, and not spontaneous DDR activation. Finally, one must bear in mind that cell cycle/proliferation can account for strong differences in the DDR, so this would need to be controlled in order to be able to draw significant conclusions.

- It is really difficult to understand how ATM inhibition can cause an increase in proliferation (Figure 5). ATM-deficient cells usually grow quite poorly, and ATM kinase dead mutants show dramatic replication defects. I am concerned that what is interpreted as more proliferation is in fact indicative of something else, such as for example an accumulation of cells in S phase due to replicative problems.

- ATM deficient tumours are frequently linked to NOTCH1 amplification. Authors should consider that ATM-linked DDR deficiency may to some degree be the cause and not the consequence of the rearrangements.

- It is also unclear why the authors measure total 53BP1 protein levels (Figure 4), what the increase in signal upon depletion of NOTCH1 means, and what the relationship with UV damage would be. 53BP1 is recruited to DNA damage sites (double-strand breaks in particular), and its recruitment to foci is widely used to estimate the number of lesions, but its overall levels are not necessarily an indicator of DDR activation.

- Regarding the NOTCH1-ATM-FOXO3A interactions (Figure 4). The rationale for these experiments is not sufficiently explained in the Results section, although it becomes clearer in the Discussion. In any case, I feel that PLA is not sufficient to claim a direct interaction that could affect ATM activity, and CoIPs would be better suited for this. Controls of the PLA including only NOTCH1 antibody are also lacking. Furthermore, authors would need to check the FOXO3A-ATM interactions in HDF, as for NOTCH1-ATM. In this regard, it is shocking that only one strain is used, and not matched with the CAFs, in line with the comment above.

- The manuscript excessively relies on the results of the laboratory regarding CSL loss that have been recently published. The connection between these two phenomena, CSL loss and NOTCH1 amplification, is still very unclear to me, and not sufficiently explored. In order to further advance in this regard, authors should aim at genetically discerning their differential contribution to HDF transformation and growth.

- The Western-blots shown to determine levels of NOTCH1 signaling (Figure 2) are not of sufficient quality. This casts doubts regarding how meaningful the differences are, especially taking into account the semiquantitative nature of analyzing Western-blots by densitometry, and the fact that no reference is made to experimental replicates.

- CGH profiles should be shown.

- Generally, the size of the Figures is too small, and it is difficult to appreciate important details.

Answers to Reviewers' comments:

Reviewer #1:

We greatly appreciate the reviewer's positive opinion and suggestions and have obtained further supporting evidence as specified below.

1. The reviewer points out a possible discrepancy between the levels of *NOTCH1* gene amplification and protein expression that we had shown for CAFs 8 and 9. We have carefully looked into this question, comparing the gene amplification and expression data among all the CAF strains included in our analysis versus patients-matched (m-HDFs) and foreskin-derived HDFs (f-HDFs). To allow a direct evaluation of levels of NOTCH1 protein expression across the various CAFs, we have repeated the immunoblot analysis of all these cells versus three f-HDF strains as outside standard control on a single immunoblot for direct comparison (Fig. 2e). As suggested by the reviewer, analysis of α SMA expression was also included. We have also repeated the quantification of *NOTCH1* gene amplification by independent analysis of the same CAFs put in culture for a second or third time (Fig. 1b, c, each dot corresponding to values obtained from independent cultures of the same strains). As shown in Fig. 1b-d, Fig. 2e and Extended Fig. 1b), there is an overall agreement between *NOTCH1* gene amplification found in CAFs relative to matched (m-HDFs) and foreskin HDFs (f-HDFs) and increased NOTCH1 expression. However, levels in increased NOTCH1 expression among various CAFs cannot be directly equated to levels of gene amplification. As we now better stress in the discussion (p. 15, line 19), this can be readily explained by the fact that increased NOTCH1 expression is likely due to several possible mechanisms besides increased copy number, including different duplication endpoints, with differential impact on distinct regulatory elements, and various other co-determining factors, including genomic alterations at other loci.

As requested, we have improved the visualization of results and added error bars to the graphs for Fig. 1b, c and Fig. 2b, c with improved statistical visualization of the results. For Figs. 1d, 2a and 4a, numbers of analyzed nuclei (n), from two cultures per strain, are now shown on top of the corresponding bars.

2. As suggested by the reviewer, we have looked in greater depth into the consequences of UVA exposure of CAFs versus m-HDFs and f-HDFs and how their response relates to NOTCH1 expression. We have devoted a whole new section of the results to this topic (p. 7, line 14), based on a combined analysis of the impact of *NOTCH1* gene silencing and activated NOTCH1 expression.

We show that acute UVA exposure caused no suppression of DNA synthesis in CAFs while this was substantially reduced in m-HDFs and f-HDFs (Fig. 3e,f). The difference can be attributed to NOTCH1, as silencing of the gene in CAFs rendered these cells sensitive to UVA-induced growth arrest without affecting m-HDFs (Fig.3e). Molecularly, we have found that *NOTCH1* silencing or inhibition in CAFs results in enhanced formation of ATM-FOXO3a complexes under basal conditions and upon UVA exposure (Fig. 5 c, d) and in activation of the downstream ATM/DDR pathway (Fig. 6a-d). The converse was found to occur in multiple f-HDF strains, in which active NOTCH1 expression suppressed

the UVA-induced formation of ATM-FOXO3a complexes (Fig. 5e) and downstream ATM signaling (Fig. 6e).

The elevated basal levels of DNA damage and γ -H2AX levels found in CAFs¹ were not further increased by *NOTCH1* silencing (Extended Fig. 2a, b). Parallel staining for apoptotic cells showed that the UVA dosage used for these experiments was not enough to induce this process, and that *NOTCH1* silencing did not increase apoptosis in CAFs or m-HDFs (Extended data Fig. 2c).

In the previous version of the paper (Fig. 3, now Fig. 4), our evidence already indicated that m-HDFs from patients contain populations with *NOTCH1* gene amplification with a selective growth advantage under conditions of persistent DNA damage resulting from repeated UVA exposures. As requested by the reviewer, we have increased the number of analyzed strains and repeated these experiments also with f-HDFs. Chronic UVA treatment caused no increase in *NOTCH1* gene copy number in multiple strains of these cells (Extended data Fig. 3c), indicating that the increase of *NOTCH1* copies in the cultures of patients derived m-HDFs is not a direct consequence of UVA treatment.

As rightfully pointed out by the Reviewer, in Figs. 3a, b and c of the previous version we did not include evaluation of statistical significance. To address this point, we have tested additional strains of CAFs, m-HDFs and f-HDFs thus allowing to add statistical calculations to the results, now shown in Fig. 3f, Extended data Fig. 3a and Fig. 4c.

We thank the reviewer for pointing out a possible connection between the accumulation of UVA skin exposure and *NOTCH1* gene amplification in dermal fibroblasts of aging skin, as we now mention in the discussion (p.15, line 23). To address this possibility experimentally will be very interesting but will require a whole new set of dedicated studies.

3. We thank the Reviewer for the suggestions on how to improve our more mechanistic/molecular findings. We performed additional PLA assays for NOTCH1-ATM and ATM-FOXO3A complex formation in CAFs versus matched m-HDFs and f-HDFs. We found low association of NOTCH1-ATM in f-HDFs and m-HDFs, while multiple complexes were detectable in CAFs already under basal conditions (Fig. 5a).

NOTCH1 binding to ATM was substantially reduced while ATM-FOXO3A association was increased in CAFs by treatment with the γ -secretase inhibitor DBZ, which suppresses NOTCH proteolytic cleavage and activation² (Fig. 5b, c; Extended Data Fig. 4b).

Importantly, ATM-FOXO3A association was induced by UVA treatment in m-HDFs but not in CAFs, except when *NOTCH1* was silenced (Fig. 5d).

Finally, increased NOTCH1 expression in multiple f-HDF strains was sufficient to block UVA induced formation of ATM-FOXO3A complexes (Fig. 5e) and ATM pathway activation (Fig. 6e).

Consistent with the above findings, we confirmed the immunofluorescence results of ATM signaling activation in CAFs with *NOTCH1* gene silencing or inhibition by immuno-blotting (Fig. 6c-d). In contrast to CAFs, phosphorylation levels of ATM, CHK2, TP53 and other downstream ATM substrates were not increased in m-HDFs by *NOTCH1* gene silencing alone, but only after UVA treatment (Extended data Fig. 4c), while phosphorylation of all these proteins in response to UVA exposure was suppressed in f-HDFs by activated NOTCH1 expression (Fig. 6e).

Regarding the elevated γ -H2AX levels noted by the reviewer in CAFs already under basal conditions, these are fully consistent with our previous report of persistent DNA damage in these cells, resulting from loss of CSL protective function at telomeres¹. The elevated basal levels of DNA damage in CAFs was also experimentally confirmed in the present study (Extended data Fig. 2 a, b; Fig. 6a, b), with values consistent with our previous publication (Fig. 2c¹).

Regarding the question on proliferation, as also explained in the text (p.7, line 19), expression of activated NOTCH1 in f-HDFs leads to CSL-dependent p21 expression and cell cycle arrest³, precluding the possibility of testing whether, in f-HDFs, increased NOTCH1 expression can overcome UVA-induced growth arrest. This is in contrast to CAFs, that are characterized by sustained NOTCH1 expression but concomitant CSL and p53/p21 down-regulation³. We show that CAFs are resistant to growth suppression by UVA treatment, unless *NOTCH1* is silenced (Fig. 3e, f).

4. A defining property of CAFs is to promote proliferation of neighboring cancer cells. In reply to the reviewer's concern that interpretation of the ear injection assays of tumor cells admixed with CAFs may be complicated by the influence of resident mouse stromal cells, we have complemented the *in vivo* results with an *in vitro* cancer / stromal cell expansion assay that we have recently developed. The assay is based on the co-culture in Matrigel of fluorescently labelled SCC cells with CAFs⁴. As shown in Fig. 8, formation of large clusters by skin SCC cells was severely reduced in the presence of multiple CAF strains with *NOTCH1* gene silencing, with similar suppressive effects being elicited by treatment of these cultures with the γ -secretase DBZ inhibitor. This compound is shown in Fig. 7e to cause no direct growth inhibition of SCC cells, which is consistent with the intrinsic tumor suppressive function of the NOTCH pathway in this cancer type⁵.

Regarding the possible influence of mouse resident stromal cells in the *in vivo* situation, we note that a similar suppression of cancer /stromal cell expansion exerted by treatment with the NOTCH inhibitor was observed with CAFs with *NOTCH1* gene silencing (Figs. 9 and 10; Extended data Fig. 6). As suggested by the reviewer, we have performed additional immunofluorescence analysis of the ear cancer lesions with human-specific versus pan-Vimentin (having reactivity for both human and mouse) antibodies. The human specific antibodies show intra- and peri-tumoral distribution of fibroblasts, while the pan-Vimentin antibodies detect also mouse fibroblast further away from the tumor area (Extended data Fig. 6c).

As for quantification of tumor formation, besides tumor cells fluorescent intensity values, we now also show tumor volumes (Figs. 9a and 10a), with results fully consistent with determination of proliferative index by Ki67 in both vimentin positive and keratin positive cells (Figs. 9b and 10b; Extended data Fig. 6a). We also specify the number of mice and ears used in treated vs control conditions for our *in vivo* experiments and indicate the number of injected ear lesions in the graphs and corresponding legends to Figs. 9a and 10a.

Reviewer #2:

We thank the reviewer for the appreciation of our work and the constructive suggestions. As recommended, we have employed a number of complementary approaches to further demonstrate that Notch favors CAF proliferation by preventing ATM signaling

downstream of γ -H2AX increase, preventing ATM-FOXO3a association and ensuing phosphorylation cascade. This was assessed by combined analysis of CAFs plus/minus *NOTCH1* silencing and HDFs plus/minus activated *NOTCH1* expression, under both basal conditions and upon UVA exposure. We have revised the paper accordingly, as indicated here below

Specific comments:

1. As requested, we have confirmed by immunoblot analysis the *NOTCH1*-dependency of ATM signaling in CAFs and UVA-treated f-HDFs and m-HDFs. Consistent with the immunofluorescence results (Fig. 6a, b), immuno-blot analysis of multiple CAF strains showed low phosphorylation levels of ATM, CHK2 and p53 and, which were all strongly induced in these cells by *NOTCH1* gene silencing or γ -secretase inhibitor treatment (Fig. 6c, d). As predicted by the reviewer, elevated γ -H2AX levels and sustained DNA damage - as detected by comet assays - were not further increased in CAFs by *NOTCH1* suppression or inhibition (Fig. 6a-d, Extended data Fig. 2a, b). In contrast to CAFs, phosphorylation levels of ATM, CHK2, p53 and γ -H2AX were not increased in m-HDFs by *NOTCH1* gene silencing alone, while they were induced by UVA treatment to a similar extent irrespectively of whether *NOTCH1* was silenced (Extended data Fig. 4c). Instead, phosphorylation of all these proteins in normal foreskin-derived HDFs (f-HDFs) upon UVA treatment was suppressed by activated *NOTCH1* expression (Fig. 6e). As requested, we have also included the total levels of proteins analyzed by immune blotting.

2. As recommended, we now show levels of *NOTCH1* expression in CAFs with shRNA-mediated gene silencing or treated with DBZ (Extended data Fig. 4a, b) as well as levels of p-S15-p53 versus total p53 in CAFs plus/minus *NOTCH1* silencing or DBZ treatment (Fig. 6c,d) and plus/minus ATM inhibition (Extended data Fig. 5d).

3. As requested, we have performed additional experiments to substantiate the effect of active *NOTCH1* in DDR signaling, DNA damage repair and proliferation blockage, showing that:

a) *NOTCH1* silencing in CAFs results in activation of the ATM/DDR pathway downstream of γ -H2AX production, by both IF and immunoblot analysis (as also indicated above) (Figure 6a-d), without further increasing the persistently elevated levels of DNA damage (Extended data Fig. 2a, b).

b) Growth suppression of CAFs by *NOTCH1* gene silencing is overcome by treatment with an ATM inhibitor at concentrations that by themselves have little effects on proliferation of these cells (Fig. 7f; Extended data Fig. 5c).

c) In contrast to CAFs, *NOTCH1* gene silencing in f-HDFs causes no growth arrest, which is instead elicited in these cells by *CSL* gene silencing, with associated induction of gene expression and DNA damage (confirming our previous results on this topic, with corresponding mechanistic analysis^{1,3,4,6} (Fig. 3 a-d).

d) Increased active *NOTCH1* expression in f-HDFs blocks the UVA-induced formation of ATM-FOXO3a complexes (Fig. 5e) and induction of the ATM/DDR pathway downstream of γ -H2AX (Fig. 6e).

As we now explain in the text (p. 7, line 19), with corresponding reference³, expression of activated *NOTCH1* in f-HDFs leads to *CSL*-dependent p21

expression and cell cycle arrest, precluding the possibility of testing whether, in f-HDFs, increased NOTCH1 expression can overcome UVA-induced growth arrest. This is in contrast with CAFs, that are characterized by sustained NOTCH1 expression and concomitant CSL down-regulation¹, in which the response to UVA is enhanced by *NOTCH1* gene silencing (Fig. 3e).

4) We thank the reviewer for raising "the possibility that UV treatment is inducing amplification of *NOTCH1* in non-amplified cells instead of expanding the *NOTCH1* amplified population". Accordingly, we have chronically treated three f-HDF strains with UVA utilizing the same conditions employed for treatment of the patients derived m-HDFs containing a small fraction of cells with *NOTCH1* gene amplification. We show that our chronic UVA treatment conditions are not sufficient to induce de novo *NOTCH1* gene amplification in f-HDFs (Extended data Fig. 3c), while they trigger an expansion of m-HDFs that carry a pre-existing *NOTCH1* gene amplification (Fig. 4 a-c).

5) Following the Reviewer's advice, we provide better images of patients derived m-HDFs plus/minus chronic UVA treatment double stained for active Notch1 and IL6, with corresponding quantification (Fig. 4b).

6) We have assessed levels of activated NOTCH1 in control f-HDFs plus/minus UVA treatment and found no modulation (Fig 6e). We further show that *NOTCH1* silencing does not affect the DDR in HDFs plus/minus UVA treatment (Extended data Fig. 4c), while activated *NOTCH1* expression in these cells is by itself sufficient to suppress activation of the ATM pathway, downstream of γ -H2AX (Fig. 6e), and to block ATM-FOXO3a association (Fig. 5e).

7) We previously reported that CAFs, as a result of decreased CSL expression, display telomere loss and fusions with persistent DNA damage and genomic instability¹. We thank the reviewer for pointing out that there are negligible levels of phospho-ATM in untreated CAFs. This is fully consistent with the previous reports that ATM auto-phosphorylation in the DDR occurs as a second step, downstream of γ -H2AX production and ATM-FOXO3A association^{7 8}. As we now show, consistent with the previously reported finding in cancer cells, suppression of *NOTCH1* expression or activity in CAFs resulted in increased ATM-FOXO3A complex formation, under both basal conditions and upon UVA exposure (Fig. 5c, d) and restored the ATM/P53 signaling cascade (Fig. 6a-d). Consistent with this mechanism of action, increased expression of activated *NOTCH1* in normal foreskin-derived HDF strains was by itself sufficient to block UVA-induced ATM-FOXO3A complex formation (Fig. 5e) and downstream ATM signaling (Fig. 6e).

Overall, as the reviewer suggests, our findings confirm that CAFs display constitutive damage in comparison with HDFs and show that increased *NOTCH1* expression and activity prevent ATM signaling, downstream of γ -H2AX. As requested, we have assessed by immunoblot analysis the ATM phosphorylation signaling cascade in CAFs plus/minus *NOTCH1* gene silencing and inhibition (Fig. 6c, d) and in f-HDFs plus/minus activated *NOTCH1* expression and concomitant UVA exposure (Fig. 6e).

This mechanism most likely applies to the *in vivo* situation as, in the orthotopic skin cancer model that we have used with CAFs plus/minus *NOTCH1* gene silencing or lesions plus/minus treatment with the γ -secretase inhibitor,

levels of phospho-ATM and downstream targets were induced by loss of NOTCH1 expression and activity (Figs. 9d and 10d).

8) A minor concern: We thank the Reviewer for pointing out that we had referred to *Bottoni et al*, as submitted, which we have now corrected.

Reviewer #3:

We thank the reviewer for the constructive suggestions and have improved the manuscript accordingly. As recommended, we provide more mechanistic insights into the connection between *NOTCH1* gene amplification and increased expression in CAFs and suppression of the ATM / DNA damage response, with parallel work performed with matched (m-HDFs) and foreskin-derived HDFs (f-HDFs). In the text (p.4, line 8), we also better explain the main novelty and significance of the findings relative to our previous work, providing additional experimental evidence in support of these conclusions. We showed before that increased NOTCH1 activity in normal fibroblasts exerts intrinsic growth suppressive effects that can be explained by conversion of CSL from a repressor to an activator of gene transcription with the concomitant induction of target genes, such as the cell cycle inhibitor *CDKN1A* and CAF effectors³. Separately from Notch activation, we more recently reported that CSL is essential for maintenance of genomic stability as part of a telomere protective complex¹. The present work bears on the important question of possible heterogeneity and genomic changes in cancer stromal fibroblasts that may contribute to cancer/stromal cell expansion. This is something that was not addressed in our previous studies; as we had already pointed out in the introduction (p. 4, line 13) and discussion (p.14, line 13), it has been a matter of contention in other systems and had never been carefully examined in the skin. Studies on genomic integrity of CAF populations in this organ are important, given the persistent exposure of the skin to exogenous clastogenic agents such as UVA, which reaches the dermal cell compartment due to its high penetrating power. In this respect, our findings of the frequent and heterogeneous levels of *NOTCH1* gene amplification in CAFs from skin SCCs, which also occur, to a lesser extent, in dermal fibroblasts of apparently unaffected skin, are an important first. Together with the underlying mechanistic insights, we provide findings of translational significance, in establishing NOTCH1 - as opposed to CSL - as an attractive target for preventing cancer / stromal cell expansion.

Specific comments:

1. We thank the Reviewer for pointing out the lack of statistical calculations for some of the experiments and information of the statistics that was used. We have increased the number of biological replicates and samples, which has allowed us to add statistical calculations where they were missing. The fact that we obtained similar results with multiple independent CAF and HDF strains (matched and foreskin-derived) is strengthening both statistical and biological significance. The statistical tests are now better specified in the corresponding methods section (p. 23, line 18) and corresponding figure legends and we have verified with a statistician that the use of a paired or unpaired t-test was appropriate.

2. As recommended, we have included additional strains of HDFs as controls. As appreciated by the reviewer, we have used matched HDFs (m-HDFs) in comparison with CAFs as much as possible. However, as we show in the paper

and consistent with the cancer fields effect, matched HDFs are also not entirely normal and contain small populations with *NOTCH1* gene amplification (Figs. 1d, 2a). For this reason, and for addressing some of the other concerns, we have also included in our analysis foreskin derived HDFs (f-HDFs) as outside controls. In fact, as requested by the reviewer, using the latter cells (multiple independent strains) we have evaluated the consequences of activated *NOTCH1* expression in UVA induction of the ATM pathway in the absence of other changes (Figs. 5e and 6e). As we explain in the text, in reference to previous work³, expression of activated *NOTCH1* in normal HDFs leads to CSL-dependent p21 expression and cell cycle arrest, precluding the possibility of testing whether, in normal HDFs, increased *NOTCH1* expression can overcome UVA-induced growth arrest. However, by focusing on UVA-induction of the ATM pathway, we now show that activated *NOTCH1* expression in f-HDFs is by itself sufficient to suppress activation of the ATM pathway downstream of γ -H2AX, blocking ATM-FOXO3A association (Fig. 5e) and ensuing phosphorylation cascade (Fig. 6e).

3. The reviewer questions why a large fraction of CAFs should be already positive for γ -H2AX in the absence of exogenous DNA damage. This was the topic of our previous publication¹ showing that elevated γ -H2AX levels in CAFs reflect persistent DNA damage in these cells, which results from loss of CSL protective function at telomeres. This was found to occur in CAFs already under basal culture conditions as well as *in vivo*. The elevated basal levels of DNA damage in CAFs were also experimentally confirmed in the present study (Extended data Fig. 2a, b).

Following the reviewer's recommendation, we have further assessed the consequences on ATM signaling of *NOTCH1* gene silencing in CAFs versus m-HDFs and f-HDFs under basal condition and in response to UVA exposure. In fact, we have devoted a whole new section of the results to this topic, based on a combined analysis of the impact of *NOTCH1* gene silencing and activated *NOTCH1* expression (p. 9, line 17).

We show that acute UVA exposure caused no suppression of DNA synthesis in CAFs while this was substantially reduced in m-HDFs and f-HDFs (Fig. 3e, f). The difference can be attributed to *NOTCH1*, as silencing of the gene in CAFs rendered these cells sensitive to UVA-induced growth arrest (Fig. 3e). Molecularly, we show that *NOTCH1* silencing or inhibition in CAFs allows formation of ATM-FOXO3A complexes (Fig. 5c, d) and activation of the downstream ATM/DDR pathway (Fig. 6a-d). The converse was found in f-HDFs, in which active *NOTCH1* expression suppressed the UVA-induced formation of ATM-FOXO3a complexes (Fig. 5e) and downstream ATM signaling (Fig. 6e).

The elevated basal levels of DNA damage and γ -H2AX found in CAFs¹ were not further increased by *NOTCH1* silencing (Extended Fig. 2a, b). Analysis of parallel cultures showed that *NOTCH1* silencing did not increase apoptosis in either HDFs or CAFs and that the UVA dosage used for these experiments was not sufficient to induce this process (Extended data Fig. 2c).

4. As suggested by the reviewer, we have looked more carefully into growth suppressing effects that may result from ATM inhibition. We note that the concentrations of the ATM inhibitor KU60019 used in our experiments are lower than those we have found to be used to trigger growth arrest in cancer cells^{9 10}. In any case, by a preliminary titration experiment in CAFs, using EdU labeling as

a read out, we chose two concentrations (500 nM and 2 μ M) that did not significantly affect proliferation of these cells (Extended data fig. 5c). By more extended time course cell density assays, which rule out S phase accumulation effects, we confirmed that the ATM inhibitor at the chosen concentration (2 μ M) did not significantly affect proliferation of multiple CAF strains under basal conditions, while it was sufficient to overcome the growth inhibitory effects resulting from *NOTCH1* gene silencing (Fig. 7f), a finding also confirmed by EdU labeling assays (Extended data fig. 5c).

5. We thank the reviewer for pointing out that it has been previously reported that ATM deficiency can be linked to aberrant double-strand DNA repair and chromosomal alterations (DSBs) leading to cancer development¹¹. Even in the present setting, ATM deficiency may not only be a consequence but to some degree a cause of *NOTCH1* gene amplifications and increased expression. This is a possibility that we are now considering in the discussion (p. 16, line 20), indicating, however, that experimentally, treatment of f-HDFs with an ATM inhibitor plus/minus prolonged UVA treatment caused no increase in *NOTCH1* gene copy number. This was instead increased in cultures of patients-derived HDFs, consistent with our other results showing the presence in these cultures of subpopulations of cells with *NOTCH1* gene amplification that are selectively expanded upon prolonged UVA treatment (Extended data Fig. 3c).

6. We thank the Reviewer for pointing out the limitations of the 53BP1 analysis. As formation of 53BP1 foci can involve a variety of complex mechanisms, we decided to focus on levels of phospho-ATM, -CHK2 and -p53 versus corresponding total proteins, as these are more directly relevant and of easier interpretation.

7. As recommended by the reviewer, in the results section, we provide a better explanation of the reasons for examining the NOTCH1/ATM and ATM/FOXO3A association (p. 9, line 18), referring to previous work with cancer cells^{7,8}. Following the reviewer's recommendation, we have included controls with anti-NOTCH1 antibodies-only in the PLA assays and performed similar assays with multiple CAF and HDF strains plus/minus *NOTCH1* gene silencing or over-expression, under basal conditions and upon UVA exposure (Fig. 5). These results show that, consistent with what reported in cancer cells^{7,8}, NOTCH1 associates with ATM and competes with FOXO3A in CAFs (Fig. 5a-d) and its increased expression in HDFs is by itself sufficient to cause these effects (Fig. 5e). Further validation by co-immunoprecipitation assays with CAFs is challenging as these cells do not grow well in culture and were already reported in previous work with cancer cells⁷.

8. As suggested by the reviewer, we have better clarified the relationship between CSL and NOTCH1 function in this system, by both referring to previous work on this topic, and experimentally.

In the text (p. 7, line 19), we indicate that : "increased NOTCH1 activity in normal fibroblasts exerts intrinsic growth suppressive effects that can be explained by conversion of CSL from a repressor to an activator of gene transcription with the concomitant induction of target genes, such as *CDKN1A* and CAF effectors³. Separately from NOTCH activation, CSL is essential for

maintenance of genomic stability as part of a telomere protective complex ¹, while activated NOTCH1 was reported, in cancer cells, to inhibit the DNA damage response by associating with the ATM/ATR kinases and suppressing the downstream signaling cascade ^{7,8}.

Experimentally, and consistent with the previous findings, we show that silencing of the *NOTCH1* gene in foreskin HDFs caused no suppression of proliferation or DNA damage (as assessed by EdU labeling, comet assays and γ -H2AX immunofluorescence), which were instead elicited by silencing of CSL, in parallel with upregulation of CSL target genes (Fig. 3a-d). By contrast, silencing of *NOTCH1* in CAFs, but not in HDFs, resulted in a significant suppression of proliferative activity under basal conditions and rendered these cells susceptible to UVA-induced growth arrest (Fig. 3e). Consistent with all our other findings and previous work ¹, CAFs exhibit high levels of DNA damage that are not further increased by *NOTCH1* gene silencing (Extended Fig. 2a, b).

9. We have improved the immunoblot analysis of NOTCH1 protein expression in CAFs versus matched and foreskin-derived HDFs. In particular, to allow a direct evaluation of NOTCH1 protein expression across the various CAF strains, we have repeated the immunoblot analysis of all these cells versus three f-HDF strains as outside controls on a single immunoblot, including analysis of α SMA expression as a CAF marker (Fig. 2e). A better exposure immunoblot of CAFs versus matched HDFs, with corresponding densitometric quantification, is shown in Extended data Fig 1b. Regarding quantification of *NOTCH1* gene amplification by FISH in Fig. 1d, the numbers of nuclei that were analyzed per strain are indicated on top of the corresponding bars.

10. The entire aCGH data profiles are provided in Supplemental Table 1 and the raw data are deposited in the public repository (GSE113577).

11. We have increased the size of figures for better visualization as suggested.

Cited literature

- 1 Bottoni, G. *et al.* CSL controls telomere maintenance and genome stability in human dermal fibroblasts. *Nat Commun* **10**, 3884, doi:10.1038/s41467-019-11785-7 (2019).
- 2 Ran, Y. *et al.* gamma-Secretase inhibitors in cancer clinical trials are pharmacologically and functionally distinct. *EMBO Mol Med* **9**, 950-966, doi:10.15252/emmm.201607265 (2017).
- 3 Procopio, M. G. *et al.* Combined CSL and p53 downregulation promotes cancer-associated fibroblast activation. *Nat Cell Biol* **17**, 1193-1204, doi:10.1038/ncb3228 (2015).
- 4 Clocchiatti, A. *et al.* Androgen receptor functions as transcriptional repressor of cancer-associated fibroblast activation. *J Clin Invest* **128**, 5531-5548, doi:10.1172/JCI99159 (2018).
- 5 Dotto, G. P. & Rustgi, A. K. Squamous Cell Cancers: A Unified Perspective on Biology and Genetics. *Cancer Cell* **29**, 622-637, doi:10.1016/j.ccell.2016.04.004 (2016).
- 6 Dong Eun Kim, M.-G. P., Soumitra Ghosh, Seung-Hee Jo, Sandro Goruppi, Francesco Magliozzi, Pino Bordignon, Victor Neel, Paolo Angelino, and Gian-

- Paolo Dotto. Convergent roles of ATF3 and CSL in chromatin control of cancer-associated fibroblasts activation. *The Journal of Experimental Medicine* (2017).
- 7 Vermezovic, J. *et al.* Notch is a direct negative regulator of the DNA-damage response. *Nat Struct Mol Biol* **22**, 417-424, doi:10.1038/nsmb.3013 (2015).
- 8 Adamowicz, M., Vermezovic, J. & d'Adda di Fagagna, F. NOTCH1 Inhibits Activation of ATM by Impairing the Formation of an ATM-FOXO3a-KAT5/Tip60 Complex. *Cell Rep* **16**, 2068-2076, doi:10.1016/j.celrep.2016.07.038 (2016).
- 9 Takeuchi, M. *et al.* Anti-Tumor Effect of Inhibition of DNA Damage Response Proteins, ATM and ATR, in Endometrial Cancer Cells. *Cancers* **11**, doi:10.3390/cancers11121913 (2019).
- 10 Golding, S. E. *et al.* Improved ATM kinase inhibitor KU-60019 radiosensitizes glioma cells, compromises insulin, AKT and ERK prosurvival signaling, and inhibits migration and invasion. *Molecular cancer therapeutics* **8**, 2894-2902, doi:10.1158/1535-7163.MCT-09-0519 (2009).
- 11 Zha, S. *et al.* ATM-deficient thymic lymphoma is associated with aberrant tcrd rearrangement and gene amplification. *J Exp Med* **207**, 1369-1380, doi:10.1084/jem.20100285 (2010).

REVIEWER COMMENTS

Reviewer #1 (Remarks to the Author):

Dear Authors,

I thank the authors for taking into consideration my comments. I found the manuscript improved. However, I would like to stress that most of the experiments have been done in a single sample, however using multiple cellular strains. Also, the use of only three mice per experimental groups seems insufficient to draw statistical conclusion to me.

A second point that I would like to rise is the lack of the molecular mechanism for tumor promotion in the context of NOTCH1 amplification in CAF. It is clear that NOTCH amplification results in CAF growth advantage. We can assume that more CAF would results in more aggressive and proliferative cancerous lesions. However, CAF can promote tumor growth through multiple modes of action (secretome, direct contact, ECM remodeling...). How NOTCH1-amplification leads to the capacity of CAF to promote cancer cells growth would have been a strength for the publication of this article.

Reviewer #2 (Remarks to the Author):

This manuscript focuses on investigating the effect of Notch amplification and activation in preventing growth arrest imposed by UV in CAFs through the inhibition of the DDR response (via competing FOXO3A/ATM interaction), which is a relevant issue. As a second review I'm used to evaluate author's response in the context of my previous comments. However, authors have here decided not to include them in the rebuttal letter, which, to my view, does not facilitate this second round of review. In any case, although I appreciate that the manuscript has been substantially improved I still found several concerns that negatively impact on the solidity of conclusions. These concerns are listed below:

1) In figure 1 b, I don't understand the selection of controls used for normalization. When looking at gene amplifications it makes no sense using a housekeeping gene as negative control (whatever it means) or GAPDH for normalization. In any case, 1.5 copies of Notch1 in CAFs 8 and 9 do not seem a substantial amplification.

2) In Figure 2 differences in Notch, hes1 and csl gene expression are minimal as well as the number of samples analyzed. Thus, concluding that CAFs display Notch1 amplification associated with increases expression is, at least, an overstatement.

3) In Figure 2d, IF analysis is insufficient (and several authors consider it a non-quantitative method) to state, "cleaved Notch1 protein is found in cultured CAFs." Moreover, in the figure it is not specified the panels corresponding to m-HDF and CAFs. Also, I do not appreciate a difference in the percent of cells with positive nuclear staining (as indicated in the graph) but a general reduction of the red intensity in the left panels. A more reliable western blot analysis of the samples (using the ICN1 antibody) with the appropriate non-overloaded controls and replicates would help to support this possibility. Just by looking at the figure, one can easily see that quantification is a little subjective. For example, CAF10 that is the one with less Notch1 is also the one showing less tubulin. In contrast, in CAF 13 the levels of tubulin are much higher (similar to Notch1).

4) In figure 3, legend is incomplete. In the comet assay, only the average value of each HDF is show, which makes difficult the interpretation of results and how statistic analysis have been applied. In addition, because CSL is required for Notch signaling and it was previously proposed that ICN1 stabilizes CSL at chromatin, it is uncertain what is the effect of Notch inhibition (or activation) in CSL levels (function) or CSL depletion in ICN1 levels and activity (should be analyzed).

5) In figure 4b, accumulation of nuclear Notch1 is massive and affects 100% of cells. This result is in contrast with the differences in gene amplification or copy number variations when comparing untreated and UV-treated cells (with at least 50% of cell with normal copy numbers in all conditions). Thus, the mechanism/s leading to Notch activation by repeated UV exposure cannot primarily be amplification of Notch1 gene. Is it possible that Notch ligands are induced by UV leading to Notch activation? Have the authors tested the levels of Jag1-2 or Dll1-4 in these CAFs?

6) The conclusion that "increased Notch activity is required for sustained proliferation in spite of persistent DNA damage" (page 9 line 12-13) is not supported by data. Similarly, the sentence "similar treatment of multiple foreskin-derived HDF strains resulted in no increased in NOTCH1 gene copy number, indicating that the increase of NOTCH1 copies is not a direct consequence of UVA treatment" have to be taken cautiously as other factors (present in CAFs but absent in f-HDF strains) can impose copy number alterations in response to UV.

7) In figure 5, results seem conclusive (in particular in 5a) but association of Notch to ATM (and its impact on FOXO3/ATM association) is only evaluated by one technique. Other experimental approaches such as Co-IP or double IF of Notch in ATM foci or BioID analysis are totally required to support the conclusion that Notch (ICN1?) interacts with ATM to prevent FOXO3-ATM interaction and DDR activation.

8) Results in figure 6 are interesting, as it seems that cells with inhibited Notch show higher levels of p-ATM, p-Chk and p-p53, however levels of active Notch1 in 6a, b, c and d should be shown. In addition, my interpretation is that the conclusion "Increased NOTCH1 expression in CAFs suppresses ATM/P53 signaling" could be perfectly changed to Notch signal inhibition in CAFs lead to increased DNA damage and DDR activation.

9) It is surprising to me that inhibition of the DDR pathway does not result in accumulation of DNA damage, thus promoting apoptosis and cell death in response to UV. This should be tested and discussed. It is also possible that intracellular Notch is sequestering free CSL thus imposing the recently demonstrated phenotype of genomic instability (Botoni et al. 2019).

10) Figures from 7 to 10 show the effect of CAFs-derived Notch activity on the growth of cancer cells. However, the mechanisms by which CAFs contribute to cancer cell growth are not addressed.

Thus, my general feeling is that whereas the manuscript clearly demonstrates that Notch activity or levels impact on CAF proliferation, UV-induced damage and DDR activation, the mechanisms underlying this effect are not properly addressed as the mechanism of Notch activation.

Reviewer #3 (Remarks to the Author):

Authors have made an important effort, substantially improving the manuscript. Many of my concerns have been addressed. Although some important issues remain, I feel that the authors will be able to deal with these problems without further experiments.

1. I still have some concerns with statistics. First, it is missing in a number of panels, and in some cases, conclusions are drawn without statistically significant differences being observed. In some particular cases, sufficient information is not provided (e.g. Fig 1c, t-test comparing what?; Fig 4c, ANOVA analyzes various things, what exactly does the p value refer to?). In others, I don't think the most appropriate test has been applied (e.g. 2-way ANOVA would be more appropriate for Fig

3e, and maybe other cases). This is particularly shocking in the case of the analysis of normalized values (e.g. Fig 2c), in which a t-test is not appropriate at all (variance in one category is artificially turned into 0). 1-sample t-test would be an acceptable alternative.

2. Based on the results in Fig 3e-f, authors claim that "unlike with HDFs, UVA treatment caused no growth suppression CAFs, unless the NOTCH1 gene was silenced". This is not true. To conclude this, authors should compare the effect of UVA treatment in siControl and siNOTCH, and not the other way around, and find statistically significant differences. If existing, these differences do not seem obvious to me.

3. The lack of effect of NOTCH1 depletion on gH2AX foci should be commented in the results section. Furthermore, I would avoid saying that phospho-ATM, -CHK2 and -p53 are low (compared to what?) and just refer to the increase observed.

4. "Relative DNA copies", in the axes in Fig 1 for example, could be misleading.

5. Information should be provided regarding the number of cells counted in PLA assays.

Reviewer #1 (Remarks to the Author):

Dear Authors,

I thank the authors for taking into consideration my comments. I found the manuscript improved.

However, I would like to stress that most of the experiments have been done in a single sample, however using multiple cellular strains. Also, the use of only three mice per experimental groups seems insufficient to draw statistical conclusion to me.

A second point that I would like to rise is the lack of the molecular mechanism for tumor promotion in the context of NOTCH1 amplification in CAF. It is clear that NOTCH amplification results in CAF growth advantage. We can assume that more CAF would results in more aggressive and proliferative cancerous lesions. However, CAF can promote tumor growth through multiple modes of action (secretome, direct contact, ECM remodeling...). How NOTCH1-amplification leads to the capacity of CAF to promote cancer cells growth would have been a strength for the publication of this article.

We thank the Reviewer for the appreciation of our work and the constructive suggestions on how to further improve it. Regarding the remaining concerns we indicate the following:

1) Multiple strains, independent repeats and statistical significance :

All experiments were based on analysis of multiple strains derived from different patients / individuals and statistical significance was calculated accordingly . In particular, CAFs and matched HDFs # 8-16 were derived from patients # 8-16; foreskin-derived HDFs #1-6 were isolated from 6 different donors. In reply to the reviewer's concern and in response to the previous recommendations, key experiments were repeated two (Fig 1b) or three (Figs 1c, 2f, 4a, 4c) independent times using the same set of strains. In addition, we note that the main conclusions are based on analysis of the same set of CAF and HDF strains by a set of complementary approaches.

Statistical significance of the findings was determined in each case across strains and across independent experimental repeats with the same strains, utilizing appropriate methods as confirmed by a statistician co-author of the paper (Dr. Paola Ostano) and as indicated in the figure legends.

Regarding the *in vivo* validation, two independent mouse experiments were performed, using in each case 4 mice with parallel ear injections of SCC cells and CAFs plus/minus *NOTCH1* gene silencing, with quantification of the results and differences that were found in all cases to be statistically significant (Fig. 9 and Extended data Fig. 6). The main findings were validated in a third independent experiment with 3 mice, with parallel ear injections plus/minus DBZ treatment, with results that were again statistically significant (Fig. 10)

2) Molecular mechanisms for tumor promotion :

We thank the reviewer for the suggestion to improve this part of the study, which we have done with more detailed explication and additional experimental support.

As we point out in the text (page. 7, line 21), a complex relationship exists between NOTCH1 and CSL activity in HDFs and CAFs. CSL functions as a constitutive negative repressor of a large battery of CAF effector genes, which

are all induced by decreased CSL expression as it occurs at early steps of CAF activation¹⁻⁴. CAF effector genes under negative CSL control can be induced by increased levels of activated NOTCH1, which, by binding to CSL, converts it from a repressor into an activator of transcription⁴. NOTCH1 activation can also suppress CSL expression as part of a negative feedback loop mediated by induction of HES/HEY family of transcriptional repressors⁵.

In agreement with previous findings^{4,6}, we show that silencing of the CSL gene in HDFs, unlike *NOTCH1*, results in upregulation of a number of CAF effector genes with a key tumor-promoting function (Fig. 3a and Extended Data Fig. 2a). Expression of all these genes was induced, and that of CSL decreased, by enhanced NOTCH1 activity, by either lentiviral-mediated expression of the activated cytoplasmic domain (ICN1) or ligand stimulation of the endogenous receptor (Fig. 3b and Extended Data Fig. 2b, d). Conversely, silencing of *NOTCH1* in CAFs caused significant downmodulation of CAF effector genes, with similar changes elicited by treatment with a γ -secretase inhibitor (DBZ) that suppresses endogenous NOTCH1 activation (Fig. 3c and Extended Data Fig. 2c). Findings were further validated by the *in vivo* tumorigenicity assays, showing even in this case, down-modulated expression of CAF effectors by *NOTCH1* gene silencing and γ -secretase treatment (Figs. 9, 10 and Extended data Fig. 6).

Thus, increased NOTCH1 activity in CAFs promotes tumorigenesis by ensuring sustained proliferation as well as the expression of a battery of CAF effector genes with established tumor promoting functions. As we point out in the discussion (page. 16, line 18), both mechanisms are interrupted by genetic or pharmacological inhibition of NOTCH1 activation in CAFs, thereby accounting for suppression of cancer / stromal cell expansion.

Reviewer #2 (Remarks to the Author):

This manuscript focuses on investigating the effect of Notch amplification and activation in preventing growth arrest imposed by UV in CAFs through the inhibition of the DDR response (via competing FOXO3A/ATM interaction), which is a relevant issue. As a second review I'm used to evaluate author's response in the context of my previous comments. However, authors have here decided not to include them in the rebuttal letter, which, to my view, does not facilitate this second round of review. In any case, although I appreciate that the manuscript has been substantially improved I still found several concerns that negatively impact on the solidity of conclusions. These concerns are listed below. Thus, my general feeling is that whereas the manuscript clearly demonstrates that Notch activity or levels impact on CAF proliferation, UV-induced damage and DDR activation, the mechanisms underlying this effect are not properly addressed as the mechanism of Notch activation.

We apologize for the lack of previous comments and we thank the reviewer for the constructive suggestions that we will address below.

1) In figure 1 b, I don't understand the selection of controls used for normalization. When looking at gene amplifications it makes no sense using a housekeeping gene as negative control (whatever it means) or GAPDH for normalization. In any case, 1.5 copies of Notch1 in CAFs 8 and 9 do not seem a substantial amplification.

Internal normalization with an unrelated chromosomal region is commonly used for determination of genomic DNA amplification by PCR assays^{7,8}. The two different genes used for normalization of our samples and a negative control, GAPDH and RPLP0 respectively, were chosen based on the fact that they didn't display genomic alterations in the aCGH array analysis, not because of their function as housekeeping genes. There was a confusing mention of "house keeping genes" in the text that has been removed.

As we point out in the figure legend, in determination of gene copy number by PCR, values with > 1.5-fold difference are generally considered as significant^{7,8}. PCR approaches provide an average quantification of a pool of cells, therefore diluting the gene amplification signal of cells that display CNVs admixed with those that carry a normal gene copy number. This is especially relevant in our case, based on analysis of heterogenous populations of freshly derived CAFs, as opposed to established cancer cells and cell lines. FISH assays provide the gold standard technique for genomic DNA copy number determination, which we have used to obviate the limitations of the PCR approach, with both freshly derived cells in culture and in tissue samples.

2) In Figure 2 differences in Notch, hes1 and csl gene expression are minimal as well as the number of samples analyzed. Thus, concluding that CAFs display Notch1 amplification associated with increases expression is, at least, an overstatement.

The relatively low number of Laser Capture Microdissection samples analyzed for levels of *NOTCH1*, *CSL* and *HES1* expression in Fig. 2c is due to the difficulties in obtaining excised SCCs and flanking normal skin tissues of sufficient quality for this type of analysis. We have analyzed one additional set of samples (CAF16 and corresponding matched fibroblasts) and, as suggested by reviewer 3, have used one sample t-test for data analysis, finding that the observed differences in *NOTCH1*, *CSL* and *HES1* expression in the SCC-associated stromal fibroblasts versus those of flanking skin, for $n(\text{matched pairs})=6$, are statistically significant (Fig. 2c). The results were complemented by similar analysis of freshly derived CAFs versus matched HDFs from these and additional three samples with similar statistically significant results, for $n(\text{matched pairs})=9$ (Fig. 2d).

3) In Figure 2d, IF analysis is insufficient (and several authors consider it a non-quantitative method) to state, "cleaved Notch1 protein is found in cultured CAFs." Moreover, in the figure it is not specified the panels corresponding to m-HDF and CAFs. Also, I do not appreciate a difference in the percent of cells with positive nuclear staining (as indicated in the graph) but a general reduction of the red intensity in the left panels. A more reliable western blot analysis of the samples (using the ICN1 antibody) with the appropriate non-overloaded controls and replicates would help to support this possibility. Just by looking at the figure, one can easily see that quantification is a little subjective. For example, CAF10 that is the one with less Notch1 is also the one showing less tubulin. In contrast, in CAF 13 the levels of tubulin are much higher (similar to Notch1).

As recommended, we have improved the IF image analysis (now Fig. 2e) and validated the results by immunoblotting (Fig. 2f). IF results with antibodies against activated NOTCH1 (ICN1) were quantified and shown as levels of NOTCH1 nuclear intensity per cell (individual dots), together with average, standard deviation and calculation of statistical significance. We have used the

same antibodies for improved immunoblot analysis of CAFs, matched HDFs as well as foreskin HDFs together with quantification of the results and assessment of statistical significance (Fig. 2f). Two other immunoblots of independent cultures of the same cells with antibodies against full length NOTCH1 are shown in Extended data Figure 1b, c.

4) In figure 3, legend is incomplete. In the comet assay, only the average value of each HDF is show, which makes difficult the interpretation of results and how statistic analysis have been applied. In addition, because CSL is required for Notch signaling and it was previously proposed that ICN1 stabilizes CSL at chromatin, it is uncertain what is the effect of Notch inhibition (or activation) in CSL levels (function) or CSL depletion in ICN1 levels and activity (should be analyzed).

We apologize for the incomplete Fig.3 legend, which has now been completed. As requested, the results of the comet assays are now shown at the level of individual cells, together with determination of statistical significance (Fig. 3d).

As indicated in the text (page. 8, line 6), NOTCH1 activation can suppress CSL expression as part of a negative feedback loop mediated by induction of HES/HEY family of transcriptional repressors⁵. As requested, we have now examined levels of CSL expression and found them to be suppressed in HDFs by increased NOTCH activity by both overexpression of activated NOTCH1 and activation of the endogenous receptor by Jagged-1 ligand stimulation (Fig. 3b and Extended Fig. 2b, d).

Regarding the consequences of CSL loss on NOTCH1 expression and activity, we show that CSL silencing in HDFs has no significant consequences on *NOTCH1* gene expression (Fig. 3a and Extended Fig. 2a). Consistent with our previous work^{2,4}, we show that expression of the *HES1* as well as *JAG1/2* ligand genes, which are commonly used as measure of NOTCH1 activity, are significantly increased in HDFs with silenced CSL (Fig. 3a). A similar induction of these genes is also observed in HDFs with increased NOTCH1 activity (Fig. 3b and Extended Fig. 2b), which can be explained by the fact that, by binding to CSL, activated NOTCH1 converts it from a repressor into an activator of transcription⁵.

5) In figure 4b, accumulation of nuclear Notch1 is massive and affects 100% of cells. This result is in contrast with the differences in gene amplification or copy number variations when comparing untreated and UV-treated cells (with at least 50% of cell with normal copy numbers in all conditions). Thus, the mechanism/s leading to Notch activation by repeated UV exposure cannot primarily be amplification of Notch1 gene. Is it possible that Notch ligands are induced by UV leading to Notch activation? Have the authors tested the levels of Jag1-2 or Dll1-4 in these CAFs?

We had showed high magnification IF images of clustered ICN1-positive cells, which did not illustrate the heterogeneity of expression. This problem has now been corrected. To test whether UVA exposure has an effect on the activation of NOTCH1, we irradiated with UVA (using the same conditions as before) multiple strains of foreskin-derived fibroblasts (that do not display pre-existing NOTCH1 gene amplification) and found that UVA treatment by itself is not sufficient to trigger NOTCH1 amplification or upregulation (Fig. 4c, d), as we now mention in the text (page. 10, line 5).

In reply to the reviewer's interesting question, we have also examined levels of NOTCH1 ligand expression in CAFs and HDFs. As we now indicate in the text

(page. 8, line 22), in many cellular systems, expression of JAGGED ligands is under positive NOTCH1 control as part of a self-reinforcing positive feedback loop⁹. We now show that JAGGED 1 and 2 expression are induced by activated NOTCH1 in f-HDFs (Fig. 3b and Extended Fig. 2b), while in CAFs, in which levels of JAGGED 1 and 2 were higher than in matched HDFs (Extended Data Fig. 2e), the expression of these genes was suppressed by *NOTCH1* silencing (Fig. 3c and Extended Fig. 2c). We further show that levels of JAGGED1 and 2 expression, like that of the *NOTCH1* gene, are unaffected or even decreased by UVA treatment at the doses used for our experiments (Extended data Fig. 3c).

6) The conclusion that “increased Notch activity is required for sustained proliferation in spite of persistent DNA damage” (page 9, line 12-13) is not supported by data. Similarly, the sentence “similar treatment of multiple foreskin-derived HDF strains resulted in no increased in NOTCH1 gene copy number, indicating that the increase of NOTCH1 copies is not a direct consequence of UVA treatment” have to be taken cautiously as other factors (present in CAFs but absent in f-HDF strains) can impose copy number alterations in response to UV.

We have modified the statements as recommended to describe more closely the findings, indicating that (page, 10, line 15) : “Thus, elevated NOTCH1 activity is required for sustained expression of CAF effector genes and CAF proliferation and *NOTCH1* gene amplification together with other factors not present in normal foreskin-derived HDFs can contribute to the CAF response to chronic UVA exposure.”.

7) In figure 5, results seem conclusive (in particular in 5a) but association of Notch to ATM (and its impact on FOXO3/ATM association) is only evaluated by one technique. Other experimental approaches such as Co-IP or double IF of Notch in ATM foci or BioID analysis are totally required to support the conclusion that Notch (ICN1?) interacts with ATM to prevent FOXO3-ATM interaction and DDR activation.

We thank the reviewer for the appreciation of our findings that we have further validated by co-immunoprecipitation assays as requested. Utilizing this approach, we confirm the exclusive association of ATM with activated NOTCH1 and not FOXO3A in multiple CAF strains; suppression of NOTCH1 activation by treatment of these cells with the γ -secretase DBZ inhibitor results in strong ATM-FOXO3A association (Fig. 5f), validating the PLA results.

8) Results in figure 6 are interesting, as it seems that cells with inhibited Notch show higher levels of p-ATM, p-Chk and p-p53, however levels of active Notch1 in 6a, b, c and d should be shown. In addition, my interpretation is that the conclusion “Increased NOTCH1 expression in CAFs suppresses ATM/P53 signaling” could be perfectly changed to Notch signal inhibition in CAFs lead to increased DNA damage and DDR activation.

As requested, we now show levels of activated NOTCH1 by immunoblot analysis to match the results shown in Fig. 6 (Fig. 6c, d and Extended data Fig. 4a, b).

We fully agree with the reviewer's alternative way to state this conclusion of our findings as indicated at the end of the relevant section of the results (page. 13, line 3) : “suppression of *NOTCH1* expression or activity in CAFs unleashes the DDR / ATM signaling cascade and TP53-dependent growth suppression”.

9) *It is surprising to me that inhibition of the DDR pathway does not result in accumulation of DNA damage, thus promoting apoptosis and cell death in response to UV. This should be tested and discussed. It is also possible that intracellular Notch is sequestering free CSL thus imposing the recently demonstrated phenotype of genomic instability (Bottoni et al. 2019).*

UVA-induced DNA damage should trigger apoptosis and cell death through activation of the DDR/p53 response, which we showed to be suppressed by increased NOTCH activity in CAFs (Figs. 6, 7). Experimentally, we have shown that *NOTCH1* silencing did not increase apoptosis in either HDFs or CAFs under basal conditions as well as after UVA exposure at the doses used for these experiments (Extended data Fig. 2h). Thus, as we point out in the discussion (page. 19, line 13), "while inhibition of the DDR/p53 signaling cascade by increased NOTCH1 activity has the potential of increasing DNA damage, the apoptotic response is also blocked and the identification of bypassing mechanisms triggering this process could be of substantial translational significance".

We thank the reviewer for raising the possibility that "*Notch is sequestering free CSL thus imposing the recently demonstrated phenotype of genomic instability*". As indicated in the discussion (page. 19, line 1), we have looked into this possibility and found that, in multiple f-HDF strains, ICN1 expression together with down-modulation of CSL expression, results in loss of CSL at telomeres and DNA damage. Given the decreased CSL levels, it is difficult to establish whether or not this protein is also sequestered out of telomeres by physical association with ICN1. Increase genomic instability in HDFs with ICN1 expression was not associated with apoptosis but growth suppression. This however can result from a second independent mechanism, involving induction of *CDKN1A* expression, a direct target of CSL transcriptional repression overcome by NOTCH activation. Because of the complexities involved, this topic warrants further biochemical investigations that are outside the scope of this paper.

10) *Figures from 7 to 10 show the effect of CAFs-derived Notch activity on the growth of cancer cells. However, the mechanisms by which CAFs contribute to cancer cell growth are not addressed.*

As we point out in the text (page. 7, line 22), CSL functions as a constitutive negative repressor of a large battery of CAF effector genes, which are all induced by decreased CSL expression as it occurs at early steps of CAF activation¹⁻⁴. CAF effector genes under negative CSL control can be induced by increased levels of activated NOTCH1, which, by binding to CSL, converts it from a repressor into an activator of transcription⁴. NOTCH1 activation can also suppress CSL expression as part of a negative feedback loop mediated by induction of HES/HEY family of transcriptional repressors⁵.

In agreement with previous findings^{4,6}, we show that silencing of the *CSL* gene in HDFs, unlike *NOTCH1*, results in upregulation of a number of CAF effector genes with a key tumor-promoting function (Fig. 3a). Expression of all these genes was induced, and that of *CSL* decreased, by enhanced NOTCH1 activity, by either lentiviral-mediated expression of the activated cytoplasmic domain (ICN1) or ligand stimulation of the endogenous receptor (Figs. 3b and Extended Data Fig. 3b). Conversely, silencing of *NOTCH1* in CAFs caused significant downmodulation of CAF effector genes, with similar changes elicited by treatment with a γ -secretase inhibitor (DBZ) that suppresses endogenous NOTCH1 activation (Figs. 3c and Extended Data Fig. 3c). Findings were further

validated by the *in vivo* tumorigenicity assays, showing even in this case, down-modulated expression of CAF effectors by *NOTCH1* gene silencing and γ -secretase treatment (Figs. 9, 10 and Extended data Fig. 6).

Thus, increased NOTCH1 activity in CAFs promotes tumorigenesis by ensuring sustained proliferation as well as expression of a battery of CAF effector genes with established tumor promoting functions. As we point out in the text (page. 15, line 1), both mechanisms are interrupted by genetic or pharmacological inhibition of NOTCH1 activation in CAFs, thereby accounting for suppression of cancer / stromal cell expansion.

Reviewer #3 (Remarks to the Author):

Authors have made an important effort, substantially improving the manuscript. Many of my concerns have been addressed. Although some important issues remain, I feel that the authors will be able to deal with these problems without further experiments.

We thank the reviewer for the appreciation of the work and have improved the paper as recommended and specified here below

1. I still have some concerns with statistics. First, it is missing in a number of panels, and in some cases, conclusions are drawn without statistically significant differences being observed. In some particular cases, sufficient information is not provided (e.g. Fig 1c, t-test comparing what?; Fig 4c, ANOVA analyzes various things, what exactly does the p value refer to?). In others, I don't think the most appropriate test has been applied (e.g. 2-way ANOVA would be more appropriate for Fig 3e, and maybe other cases). This is particularly shocking in the case of the analysis of normalized values (e.g. Fig 2c), in which a t-test is not appropriate at all (variance in one category is artificially turned into 0). 1-sample t-test would be an acceptable alternative.

We apologize for the lack of statistical analysis for some of the figures and incomplete information. We have performed calculations of statistical significance in all panels in which these were missing and verified with a biostatistician, who is a co-author of the paper (Dr. Paola Ostano), that the adopted methods were, in each case, appropriate. We thank the reviewer for recommending use of 1-sample t-test for the data of Fig. 2c, with differences which we have found to be statistically significant.

2. Based on the results in Fig 3e-f (now Fig. 3g-h), authors claim that "unlike with HDFs, UVA treatment caused no growth suppression CAFs, unless the NOTCH1 gene was silenced". This is not true. To conclude this, authors should compare the effect of UVA treatment in siControl and siNOTCH, and not the other way around, and find statistically significant differences. If existing, these differences do not seem obvious to me.

We apologize for the confusing presentation of the data, which we have now corrected. We now first show the comparison of various HDF and CAF strains plus/minus UVA treatment. In contrast to foreskin-derived and matched-HDFs, proliferation of CAFs is unaffected by UVA treatment (Fig. 3h). We then show that proliferation of CAFs is suppressed by *NOTCH1* silencing, while that of HDFs is unaffected (Fig. 3g, left columns). Finally, we show that inhibition of

CAF proliferation by *NOTCH1* silencing is not further decreased by UVA treatment (Fig. 3g, right columns).

We note that the results are fully consistent with our other findings that activated *NOTCH1* suppresses the DDR, under basal conditions and upon UVA exposure, at the level ATM-FOXO3A association and downstream events (Fig. 6).

3. *The lack of effect of NOTCH1 depletion on γH2AX foci should be commented in the results section. Furthermore, I would avoid saying that phospho-ATM, -CHK2 and -p53 are low (compared to what?) and just refer to the increase observed.*

We have modified the text as recommended (page. 11, line 22) indicating that "immunofluorescence and immunoblot analysis of multiple CAF strains showed that γ-H2AX levels were not significantly affected by *NOTCH1* silencing, while phosphorylation levels of ATM, CHK2, p53 and other downstream ATM substrates (as detected by anti-pS/TQ antibodies), were all strongly induced by *NOTCH1* gene silencing or γ-secretase inhibitor treatment (Fig. 6a-d)".

4. *"Relative DNA copies", in the axes in Fig 1 for example, could be misleading.*

As recommend, we have changed "Relative DNA copies", in the axes in Fig1. b-c, Fig. 2b, Fig. 4b-c, Extended data Fig. 3b : "relative copy number" as indicated for similar measurements in other papers ^{7,8}.

5. *Information should be provided regarding the number of cells counted in PLA assays.*

The number of cells counted in the PLA assays was already provided in the previous version of the paper are now referred to as "n(cells) =" in the Fig 5a-e legends.

Cited literature

- 1 Clocchiatti, A. *et al.* Androgen receptor functions as transcriptional repressor of cancer-associated fibroblast activation. *J Clin Invest* **128**, 5531-5548, doi:10.1172/JCI99159 (2018).
- 2 Kim, D. E. *et al.* Convergent roles of ATF3 and CSL in chromatin control of cancer-associated fibroblast activation. *J Exp Med* **214**, 2349-2368, doi:10.1084/jem.20170724 (2017).
- 3 Menietti, E. *et al.* Negative control of CSL gene transcription by stress/DNA damage response and p53. *Cell cycle* **10**, 1-12, doi:10.1080/15384101.2016.1186317 (2016).
- 4 Procopio, M. G. *et al.* Combined CSL and p53 downregulation promotes cancer-associated fibroblast activation. *Nat Cell Biol* **17**, 1193-1204, doi:10.1038/ncb3228 (2015).
- 5 Kopan, R. & Ilagan, M. X. The canonical Notch signaling pathway: unfolding the activation mechanism. *Cell* **137**, 216-233, doi:S0092-8674(09)00382-1 [pii] 10.1016/j.cell.2009.03.045 (2009).
- 6 Bottoni, G. *et al.* CSL controls telomere maintenance and genome stability in human dermal fibroblasts. *Nat Commun* **10**, 3884, doi:10.1038/s41467-019-11785-7 (2019).
- 7 Goh, J. Y. *et al.* Chromosome 1q21.3 amplification is a trackable biomarker and actionable target for breast cancer recurrence. *Nat Med* **23**, 1319-1330, doi:10.1038/nm.4405 (2017).

- 8 Ma, L. & Chung, W. K. Quantitative analysis of copy number variants based on real-time LightCycler PCR. *Curr Protoc Hum Genet* **80**, Unit 7 21, doi:10.1002/0471142905.hg0721s80 (2014).
- 9 Bray, S. J. Notch signalling: a simple pathway becomes complex. *Nat Rev Mol Cell Biol* **7**, 678-689 (2006).

REVIEWERS' COMMENTS:

Reviewer #1 (Remarks to the Author):

I thank the authors for taking into consideration my comments. I have no further concern regarding this manuscript.

Reviewer #2 (Remarks to the Author):

The authors have now addressed all my previous concerns.

Reviewer #3 (Remarks to the Author):

The authors have done an important effort in clarifying the issues raised by the referees. I am now overall satisfied with their treatment of the data. I still have, however a concern with current Figure 3g. My interpretation of this set of data, now with more thorough statistical analysis, is that clearly NOTCH1 confers a proliferative advantage to CAFs, and that CAFs are resistant to UV, but I fail to see any evidence of these phenotypes being linked. This is, one would expect that in conditions of silenced NOTCH1, CAFs would become sensitive to UV, and this is not the case (note non-statistically significant differences between columns 5-6 and 11-12). I am sorry to insist, but I consider essential to clarify this.

Reviewer #3 (Remarks to the Author):

The authors have done an important effort in clarifying the issues raised by the referees. I am now overall satisfied with their treatment of the data. I still have, however a concern with current Figure 3g. My interpretation of this set of data, now with more thorough statistical analysis, is that clearly NOTCH1 confers a proliferative advantage to CAFs, and that CAFs are resistant to UV, but I fail to see any evidence of these phenotypes being linked. This is, one would expect that in conditions of silenced NOTCH1, CAFs would become sensitive to UV, and this is not the case (note non-statistically significant differences between columns 5-6 and 11-12). I am sorry to insist, but I consider essential to clarify this.

Answer

We thank the reviewer for the further suggestion to improve this part of the study, which we have done as explained below.

In the experiment of Fig. 3 g, h, based on siRNA-mediated silencing of the *NOTCH1* gene, we had shown that *NOTCH1* knockdown suppressed the proliferation of CAFs (Fig. 3g, left columns) and that, in contrast to foreskin-derived and matched-HDFs, proliferation of CAFs was unaffected by UVA treatment (Fig. 3g, h). As the reviewer points out, UVA treatment of CAFs with *NOTCH1* gene silencing resulted in limited further suppression of proliferation. This was at the borderline of statistical significance ($p < 0.07$), as it occurred in only two of the three tested CAF strains. As shown in Fig. 3i, UVA treatment of the same CAF strains with silencing of the *NOTCH1* gene with a more prolonged period of time by an shRNA-mediated approach resulted in a more consistent and statistical significant suppression of proliferation in all three strains, in keeping with the protective role of increased NOTCH1 expression in CAFs.